# Novel mechanistic insights into the role of Mer2 as the keystone of meiotic DNA break formation

**Dorota Rousová[1], Vaishnavi Nivsarkar[2†], Veronika Altmannova[1†], Vivek B Raina[2,3†], Saskia K Funk[1], David Liedtke[1], Petra Janning[4], Franziska Müller[2], Heidi Reichle[1], Gerben Vader[2,5], John R Weir[1]\***

[1]Friedrich Miescher Laboratory of the Max Planck Society, Tübingen, Germany; [2]Department of Mechanistic Cell Biology, Max Planck Institute of Molecular Physiology, Dortmund, Germany; [3]Columbia University Medical Center, New York, United States; [4]Department of Chemical Biology, Max Planck Institute of Molecular Physiology, Dortmund, Germany; [5]Section of Oncogenetics, Department of Human Genetics, Cancer Centre Amsterdam and Amsterdam Reproduction and Development Research Institute, Amsterdam, Netherlands

**Abstract** In meiosis, DNA double-strand break (DSB) formation by Spo11 initiates recombination and enables chromosome segregation. Numerous factors are required for Spo11 activity, and couple the DSB machinery to the development of a meiosis-specific 'axis-tethered loop' chromosome organisation. Through in vitro reconstitution and budding yeast genetics, we here provide architectural insight into the DSB machinery by focussing on a foundational DSB factor, Mer2. We characterise the interaction of Mer2 with the histone reader Spp1, and show that Mer2 directly associates with nucleosomes, likely highlighting a contribution of Mer2 to tethering DSB factors to chromatin. We reveal the biochemical basis of Mer2 association with Hop1, a HORMA domain-containing chromosomal axis factor. Finally, we identify a conserved region within Mer2 crucial for DSB activity, and show that this region of Mer2 interacts with the DSB factor Mre11. In combination with previous work, we establish Mer2 as a keystone of the DSB machinery by bridging key protein complexes involved in the initiation of meiotic recombination.

**\*For correspondence:**
john.weir@tuebingen.mpg.de

†These authors contributed equally to this work

## Editor's evaluation

Using a combination of biochemical approaches and yeast genetics, the authors study the function of the DNA double-strand break factor Mer2. Rousova et al., show that Mer2 interacts with a meiotic chromosome axis factor (Hop1), nucleosomes, the nucleosome-binding protein Spp1, and the double-strand break factor Mre11 to serve as a "keystone" for meiotic DNA break formation. These findings represent an important step forward in understanding the functions of this highly conserved protein in meiosis.

## Introduction

Meiotic recombination is one of the defining features of eukaryotic sexual reproduction. In addition to creating the genetic diversity that fuels speciation and evolution, meiotic recombination fulfils a direct mechanistic role in establishing connections between initially unpaired homologous chromosomes. Meiotic recombination is initiated by programmed DNA double-strand break (DSB) formation by the transesterase Spo11 (*Keeney et al., 1997*). Meiotic DSBs are preferentially repaired via

**eLife digest** Organisms are said to be diploid when they carry two copies of each chromosome in their cells, one from each of their biological parents. But in order for each parent to only pass on one copy of their own chromosomes, they need to make haploid cells, which only carry one copy of each chromosome. These cells form by a special kind of cell division called meiosis, in which the two chromosomes from each pair in the parent cells are first linked, and then pulled apart into the daughter cells.

Accurate meiosis requires a type of DNA damage called double-stranded DNA breaks. These breaks cut through the chromosomes and can be dangerous to the cell if they are not repaired correctly. During meiosis, a set of proteins gather around the chromosomes to ensure the cuts happen in the right place and to repair the damage. One of these proteins is called Mer2. Previous studies suggest that this protein plays a role in placing the DNA breaks and controlling when they happen.

To find out more, Rousova et al. examined Mer2 and the proteins that interact with it in budding yeast cells. This involved taking the proteins out of the cell to get a closer look. The experiments showed that Mer2 sticks directly to the chromosomes and acts as a tether for other proteins. It collaborates with two partners, called Hop1 and Mre11, to make sure that DNA breaks happen safely. These proteins detect the state of the chromosome and repair the damage. Stopping Mer2 from interacting with Mre11 prevented DNA breaks from forming in budding yeast cells.

Although Rousova et al. used budding yeast to study the proteins involved in meiosis, similar proteins exist in plant and animal cells too. Understanding how they work could open new avenues of research into cell division. For example, studies on plant proteins could provide tools for creating new crop strains. Studies on human proteins could also provide insights into fertility problems and cancer.

recombination from the homologous chromosome which, depending on how recombination intermediates are processed, can yield crossovers (reviewed in *Hunter, 2015*). Together with sister chromatid cohesion, crossovers provide the physical linkage between homologous chromosomes which is necessary to ensure meiotic faithful chromosome segregation. In most organisms, crossover formation is associated with, and influenced by synapsis between homologs, established by the assembly of the SC. The formation of meiotic DSBs by Spo11 needs to be carefully orchestrated and controlled. In addition to Spo11, at least 10 additional factors are required for Spo11-dependent DSB activity, and collectively these factors are referred to as the meiotic DSB machinery. Functional and biochemical analysis has begun to reveal the logic of the assembly of the DSB machinery. A picture is emerging in which several distinct subcomplexes are co-recruited into Spo11 activity proficient chromosomal foci. In addition to the core DSB machinery, several other factors promote meiotic DSB activity. For example, DSB formation occurs in the context of a distinctive chromatin loop axis architecture which is formed concomitantly with the entry of cells into the meiotic program (*Figure 1A*). In budding yeast, this proteinaceous axis is made up of a meiosis-specific cohesin complex (Rec8 cohesin) in combination with the coiled-coil scaffolding protein Red1 and the HORMA domain protein Hop1 (*Smith and Roeder, 1997*). In cells lacking meiotic axis components, Spo11-dependent DSB formation is severely impaired (but not completely abolished, as in DSB machinery mutants), and efficient recruitment of meiotic DSB factors depends on axis establishment. In addition, DSB placement and formation is influenced by histone modifications (specifically histone H3-K4 methylation, which in budding yeast cells directs Spo11 to gene promotor regions). These nucleosomal interactions of the DSB machinery are proposed to occur within genomic regions which are located in the chromatin loops that emanate away from the chromosome axis (to which the DSB machinery is tethered).

A key component of the meiotic DSB machinery is Mer2. This protein (also known as Rec107) was originally identified as a high-copy number suppressor of the *mer1* phenotype (Mer1 was later shown to be a cofactor for the splicing of various meiotic mRNAs, including Mer2; *Engebrecht et al., 1991*), after which it was shown to be essential for meiosis (*Engebrecht et al., 1990*). Mer2 is central to the temporal control of Spo11-dependent DSB formation, being the target of S-Cdk and DDK (Cdc7-Dbf4) phosphorylation that presumably allows the binding of the Spo11-associated factors Rec114 and Mei4 to Mer2 (*Matos et al., 2008*; *Murakami and Keeney, 2014*; *Wan et al., 2008*). This regulation plays a crucial role in the spatiotemporal assembly of the DSB machinery. In addition

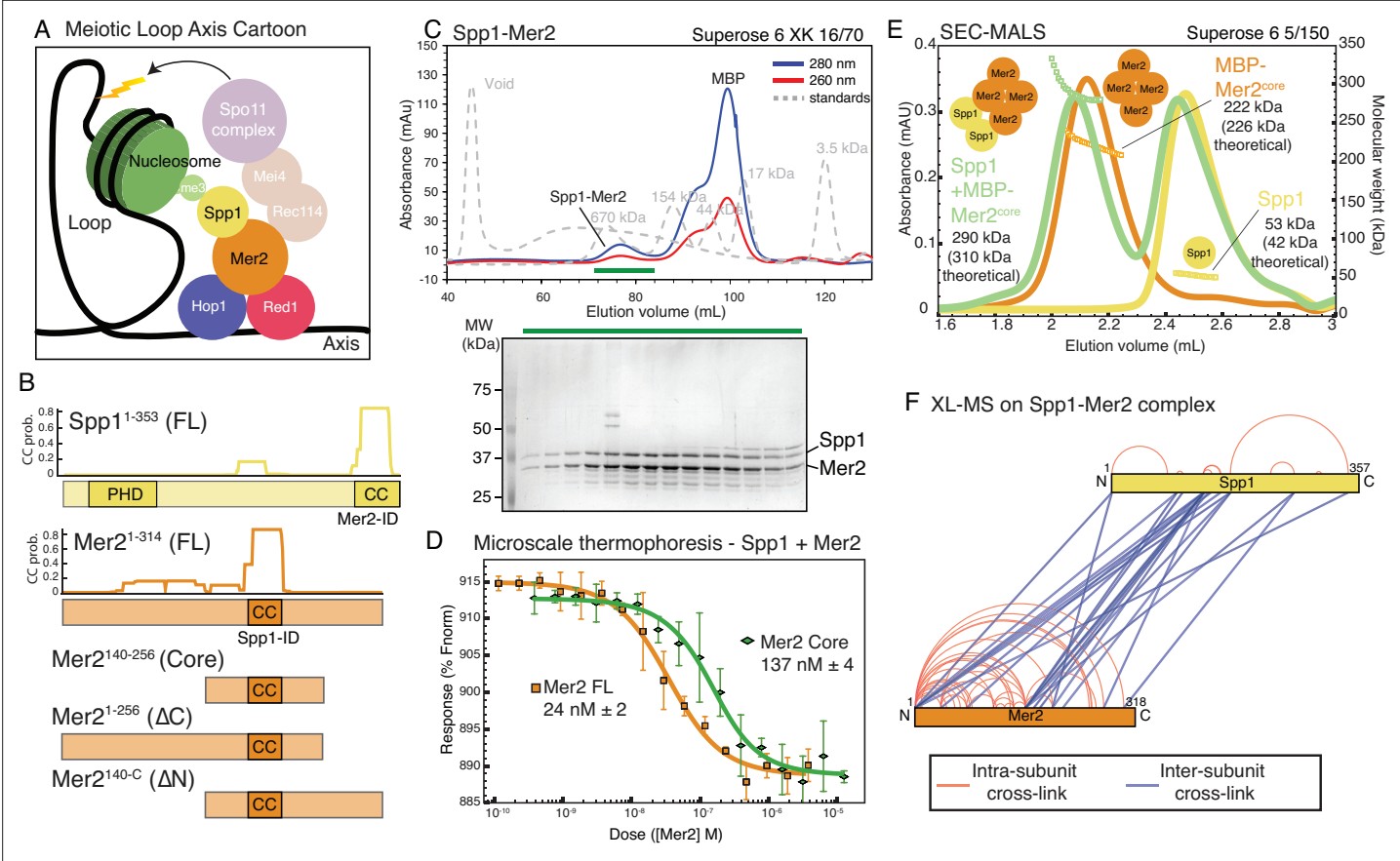

**Figure 1.** Spp1 binds to Mer2 tetramerisation domain with a 2:4 stoichiometry. (**A**) Cartoon of meiotic loop axis architecture and the role of Mer2. During meiosis proteins Red1 and Hop1 form a protein:DNA (coloured black) axis together with cohesin (not shown for clarity). Loops of chromatin are extruded from the axis, where DNA breaks are made by the Spo11 complex (pale magenta). Breaks are directed to the proximity of H3K4me3 nucleosomes (green) through the combined activity of Mer2 and Spp1. Mer2 (orange) is also thought to interact with additional Spo11 accessory proteins (Rec114 and Mei4, pale orange). (**B**) Domain diagram of Mer2 and Spp1. The four principal Mer2 constructs used throughout this study are shown. The only clear feature of Mer2 (and its orthologs IHO1 in mammals, Rec15 in fission yeast, and Prd3 in plants) is the central coiled-coil motif. Spp1 is shown for comparison with its N-terminal PHD domain and C-terminal Mer2 interaction domain which is predicted to contain a coiled-coil. Predicted coiled-coil probability (CC prob.) is shown for Spp1 full-length and Mer2 and is based on a 28 amino acid window using PCOILS (*Gruber et al., 2006*). (**C**) Purification of Mer2-Spp1 complex. A complex of Mer2 and Spp1 was purified to homogeneity. The MBP tags on Spp1 and on Mer2 were cleaved prior to loading. Degradation products of Mer2 full-length protein are also visible. Molecular weight markers are shown in grey. The relative absorbance of the complex at 280 and 260 nm shows that it is free of any significant nucleic acid contamination. The selected fractions (green line) were loaded onto an SDS-PAGE gel and stained with InstantBlue. (**D**) Microscale thermophoresis of Mer2-Spp1. Two different Mer2 constructs (Mer2[FL], orange squares; Mer2[core] green diamonds) were titrated against Red-NHS labelled untagged Spp1 (20 nM constant concentration), and the change in thermophoresis was measured. Experiments were carried out in triplicate and the $K_D$ was determined from the fitting curve. (**E**) Size exclusion chromatography coupled to multi-angle light scattering (SEC-MALS) of different Mer2-Spp1 samples. Measured molecular masses are indicated Three illustrative SEC-MALS experiments are shown for Spp1 (yellow) (theoretical mass of monomer = 42 kDa), MBP-Mer2[core] (orange) (theoretical mass of a tetramer = 226 kDa) and a complex of Spp1 bound to MBP-Mer2[core] (green) (theoretical mass of 2:4 complex = 310 kDa). (**F**) XL-MS of Mer2-Spp1 complex. Full-length Spp1 and Mer2 were used, and in both cases the overhang remaining after cleaving the N-terminal fusion proteins was present. Samples were cross-linked with disuccinimidyl dibutyric urea (DSBU), and data analysis carried out according to the Materials and methods. Cross-links were filtered so as to give a 1% false discovery rate. Figure was prepared using XVis (*Grimm et al., 2015*). Intramolecular cross-links shown in red, intermolecular cross-links in blue.

The online version of this article includes the following source data and figure supplement(s) for figure 1:

**Source data 1.** Raw gel data for *Figure 1* size exclusion chromatography (SEC) profile.

**Figure supplement 1.** Microscale thermophoresis (MST) measurements on additional Mer2 constructs.

**Figure supplement 2.** Interaction regions of Spp1 with Mer2.

**Figure supplement 2—source data 1.** Raw gel data for *Figure 1—figure supplement 2* pulldowns.

**Figure supplement 3.** Additional size exclusion chromatography coupled to multi-angle light scattering (SEC-MALS) chromatographs and summary.

**Figure supplement 4.** Cross-linking coupled to mass spectrometry (XL-MS) on Mer2 alone.

to forming a complex with Rec114-Mei4, Mer2 interacts directly with the PHD domain-containing protein Spp1 (*Acquaviva et al., 2013*; *Sommermeyer et al., 2013*). Spp1 binds to nucleosomes that are tri- (or di-) methylated on H3K4 (i.e. H3K4$^{me3}$ nucleosomes) (*He et al., 2019*; *Miller et al., 2001*), and this association is important for the association of Spo11-dependent DSB formation with gene promoter regions. Spp1 is canonically part of the COMPASS (aka Set1 complex), but during meiosis, Spp1 forms an independent, and mutually exclusive, interaction with Mer2. The reciprocal interaction domains between Spp1 and Mer2 have been previously identified (*Acquaviva et al., 2013*; *Sommermeyer et al., 2013*). The C-terminal region of Spp1 interacts with a central, predicted coiled-coil, region of Mer2 (*Acquaviva et al., 2013*; *Sommermeyer et al., 2013*; *Figure 1B*). A single amino acid substitution in Mer2 (V195D) is sufficient to disrupt the interaction with Spp1 (as judged by yeast-2-hybrid [Y2H] analysis) (*Adam et al., 2018*). Through its interaction with Spp1, a key role for Mer2 appears to link the Spo11 machinery directly to Spp1-mediated nucleosome interactions. Interestingly, Spp1 associated with Mer2 has a longer residence time on nucleosomes when compared with Spp1 when part of COMPASS (*Karányi et al., 2018*), suggesting additional functions for Mer2 in mediating nucleosome tethering. In line with the central position for Mer2 in DSB machinery assembly is the observation that – in contrast to deletion of Set1 or Spp1 which severely reduces, but not eliminates DSB formation – *mer2Δ* cells completely fail to form meiotic DSBs (*Rockmill et al., 1995*). Mer2 likely establishes additional biochemical interactions that enable a functional Spo11 assembly. For example, homologs of Mer2 in fission yeast (*Kariyazono et al., 2019*) and mouse (*Stanzione et al., 2016*) interact with meiotic chromosome axis-associated HORMA proteins, suggesting that Mer2 can mediate a link between the chromosome axis (via HORMA protein interaction) and chromatin loops (through Spp1 association).

Despite hints to the central position of Mer2 in assembly of DSB machinery, a more comprehensive biochemical understanding of these interactions is critically needed. Here, we use a combination of in vitro biochemical reconstitution with yeast genetics to investigate several distinct protein-protein interactions of Mer2. We examine the interaction of Mer2 with Spp1, nucleosomes, with proteins of the meiotic axis, and with additional members of the DSB machinery. Our results report a more complete picture of Mer2 as a foundational component of the meiotic DSB machinery, including novel functions, and provide mechanistic explanations for a number of previously observed phenomena revolving around the regulation of meiotic DSB formation.

## Results

### Mer2-Spp1 is a complex with a 4:2 stoichiometry

We first focussed on the described interaction between Mer2 and Spp1. In order to probe the various possible functions of Mer2, we made use of four principal expression constructs, the full-length protein (residues 1–314 from hereon abbreviated to 'Mer2$^{FL}$'), Mer2 amino acids 1–256, lacking the C-terminal 58 residues (from hereon 'Mer2$^{\Delta C}$'), Mer2 residues 140–314 (i.e. lacking the N-terminal 139 residues; from hereon 'Mer2$^{\Delta N}$') and Mer2 containing residues 140–256 (from hereon referred to as 'Mer2$^{core}$') and a putative coiled-coil domain (*Figure 1B*). Previous work has identified the Spp1 interaction region to be contained within Mer2$^{core}$ (specifically Mer2 residues 165–232; *Acquaviva et al., 2013*). Using our in-house expression system, 'InteBac'(*Altmannova et al., 2021*), we produced full-length Spp1 and all Mer2 proteins in *Escherichia coli* with N-terminal MBP tags to facilitate protein solubility. We could successfully remove the MBP tag using the 3C protease from both Spp1 and Mer2, though in the case of Mer2$^{\Delta N}$ and Mer2$^{Core}$ the 3C cleavable MBP tag could not be removed, presumably due to steric hindrance by the MBP tag that precludes efficient cleavage. In co-lysis experiments, we found that we could purify a complex of Mer2 and Spp1 to homogeneity, and free of nucleic acid contamination (Mer2$^{FL}$ with Spp1 shown as an example in *Figure 1C*, note the apparent A$^{260}$ to A$^{280}$ ratio as evidence of a lack of nucleic acid contamination), thus indicating that the interaction between Mer2 and Spp1 does not require any PTMs or additional cofactors and that the interaction was robust enough to survive extensive co-purification in >300 mM NaCl.

Using microscale thermophoresis (MST) we measured the binding affinity of Spp1 to Mer2$^{FL}$ (*Figure 1D* orange trace, squares) and Spp1 to Mer2$^{Core}$ (*Figure 1D* green trace, diamonds). Spp1 bound Mer2$^{FL}$ with a $K_D$ of 24 nM (±2), and to Mer2$^{Core}$ with a $K_D$ of 137 nM (±4). Mer2 constructs lacking the 'core' showed comparatively weak binding (*Figure 1—figure supplement 1*). Thus, we

confirm that the majority of the Spp1 binding interface is indeed within the core of Mer2, as reported earlier (*Acquaviva et al., 2013*; *Adam et al., 2018*), although there does appear to be some contribution to Spp1 binding provided by the N- and C-terminal regions of Mer2. We next tested the reciprocal interaction using a C-terminally 2xStrep-II-tagged Mer2$^{FL}$ against full-length Spp1, and two additional Spp1 constructs, Spp1$^{ΔC}$, containing amino acids 1–170, and Spp1$^{ΔPHD}$ containing amino acids 169–353. We found that in a Strep-Mer2$^{FL}$ pulldown on Streptactin beads Spp1$^{FL}$ and Spp1$^{ΔPHD}$ interacted with Mer2, but Spp1$^{ΔC}$ did not, consistent with previous studies (*Figure 1—figure supplement 2*).

We measured the molecular mass of Mer2 by size exclusion chromatography coupled to multi-angle light scattering (SEC-MALS) and concluded that Mer2$^{core}$ is the tetramerisation region (*Figure 1E*, orange trace and *Figure 1—figure supplement 3A-D*), consistent with recent observations of Mer2 (*Claeys Bouuaert et al., 2021*). Interestingly, while Mer2$^{ΔCΔcore}$ is monomeric (*Figure 1—figure supplement 3D*), Mer2$^{ΔNΔcore}$ is dimeric (*Figure 1—figure supplement 3C*) indicating the presence of a dimerisation region in the C-terminal region of Mer2 between residues 255 and 314, which presumably aids in the stability of a full coiled-coil tetramer.

Given that the tetramerisation region of Mer2 is also the principal Spp1 binding region (*Figure 1D*), we determined the stoichiometry of the Mer2-Spp1 complex. First, we determined that full-length Spp1 alone is a monomer (*Figure 1E* yellow trace, *Figure 1—figure supplement 3E-F*). We next analysed the stoichiometry of Mer2:Spp1 complexes. We measured the size of a complex of MBP-Mer2$^{core}$ with Spp1 (*Figure 1E* green trace) and determined its mass to be 290 kDa. The theoretical mass of a 4:2 (MBP-Mer2$^{core}$:Spp1) complex is 310 kDa, whereas a 4:1 (MBP-Mer2$^{core}$:Spp1) complex is 268 kDa. Given the possible ambiguity in determination of stoichiometry, we also measured complexes of MBP-Mer2$^{FL}$ and untagged Mer2$^{FL}$ together with Spp1 (*Figure 1—figure supplement 3G* and H) which gave complex sizes best fitting a 4:2 (Mer2:Spp1) stoichiometry. Taken together we conclude that the Mer2 tetramer binds two copies of Spp1, establishing a complex in a 4:2 stoichiometry. Thus we show a novel function for Mer2 in not simply binding Spp1, but importantly, in mediating the dimerisation of Spp1. In light of the inherent 2-fold symmetry of nucleosomes, we suggest that this 4:2 constellation might aid in the recognition of modified nucleosome tails by Spp1.

Next we probed the structural organisation of Mer2-Spp1 further using cross-linking coupled to mass spectrometry (XL-MS) using the 11 Å spacer crosslinker disuccinimidyl dibutyric urea (DSBU) (*Figure 1F*). XL-MS of Mer2-Spp1 revealed that, while the 'core' of Mer2 showed many cross-links with Spp1, these were, unexpectedly, not with the previously described C-terminal interaction domain 'Mer2-ID' of Spp1 (*Acquaviva et al., 2013*). Instead the Mer2 core showed numerous cross-links with a region of Spp1 immediately C-terminal to the PHD domain. Furthermore there were additional cross-links between Spp1 and the N- and C-terminal regions of Mer2, consistent with the residual binding affinity we observed in MST (*Figure 1—figure supplement 1*). The intramolecular cross-linked pattern of Mer2 alone (*Figure 1—figure supplement 4A*) was very similar in the presence and absence of Spp1. As such we can possibly exclude a significant structural rearrangement of Mer2 upon association with Spp1. We also compared the cross-linking pattern observed previously for Mer2 alone (*Claeys Bouuaert et al., 2021*; *Figure 1—figure supplement 4B*). This revealed that the pattern was broadly similar with a mixture of long- and short-distance cross-links. One striking difference is the extensive cross-links emanating from the N-term of our Mer2. The most likely explanation is that the overhang remaining on our Mer2 preparation after removal of the N-terminal fusion protein is four amino acids longer, and thus more flexible.

## Mer2-Spp1 complex binding to H3K4me3 modified mononucleosomes

In order to study the role of the Mer2-Spp1 complex binding to H3K4$^{me3}$ nucleosomes, we created synthetic H3K4$^{me3}$ mononucleosomes. Briefly, we mutated K4 of histone H3 to cysteine whereas the single naturally occurring cysteine of the natural H3 sequence was mutated to alanine (C110A). H3C4 was converted to H3K4$^{me3}$ using a trimethyllysine analogue as previously described (*Simon et al., 2007*). We then reconstituted H3K4$^{me3}$ into octamers, and subsequently into mononucleosomes using 167 bp Widom sequences (see Materials and methods for further details). Due to the dimeric nature of nucleosomes, and because our reconstitutions showed a 4:2 Mer2:Spp1 complex stoichiometry, we hypothesised that dimerisation of Spp1 might lead to more tight binding to H3K4$^{me3}$ nucleosomes, and set out to test this idea. We compared the apparent nucleosome binding affinity

of Spp1, GST-tagged Spp1 (which mediates dimerisation), and the Mer2-Spp1 complex using both electrophoretic mobility shift assays (EMSAs) (*Figure 2A*) and pulldowns on biotinylated nucleosomes (*Figure 2B*). We observed that the Mer2-Spp1 assembly (in which two copies of Spp1 are present) bound more tightly to nucleosomes, as compared to monomeric Spp1 alone (which bound relatively weakly to nucleosomes, consistent with the reported ~1 μM affinity of the PHD domain with H3K4$^{me3}$ peptide; *He et al., 2019*). Interestingly, when we compared the observed binding of Spp1:Mer2 with GST-Spp1, we found that GST-Spp1 exhibited an intermediate apparent binding affinity. A potential corollary of this observation is that Mer2 in addition to triggering Spp1 'dimerisation' might directly contribute to nucleosome binding.

We established that the Mer2-Spp1 complex was capable of forming a stable complex with H3K4$^{me3}$ nucleosomes in solution using analytical SEC (*Figure 2C*). Note that we do not observe a complete shift of nucleosomes; we suspect that this is due to not having an optimal buffer condition (in this experiment we have tried to balance the buffer conditions required for Mer2 [high salt], Spp1 [Zn$^{2+}$ ions for the PHD domain], and nucleosomes [EDTA]). As our complex is currently not suitable for high-resolution structural studies, we made use again of XL-MS to determine a topological architecture of the Mer2-Spp1-H3K4$^{me3}$ mononucleosome complex (*Figure 2D*). We detected many more internal Spp1 cross-links than observed in the Mer2-Spp1 complex alone (*Figure 1F*), suggesting that either the binding to nucleosomes brings the two Spp1 moieties (in the 2:4 complex) closer to one another, or that there is an internal rearrangement of domains of Spp1. Most strikingly however, the cross-linking revealed that most of the cross-links between the Mer2-Spp1 complex and the nucleosomes are via regions of Mer2, in the N-term, core, and C-term regions. This observation strengthened the idea that Mer2 might directly contribute to nucleosome binding. We modelled the location of the Mer2-Spp1 cross-links onto the previously determined structure of a mononucleosome (*Davey et al., 2002*; *Figure 2E*). We observe that the cross-links cluster around histone H3, but more generally around the DNA entry/exit site on the nucleosomes. These observations suggest a large Mer2:nucleosome interface, and a considerably smaller Spp1:nucleosome interface.

## Mer2 binds directly to nucleosomes with a 4:1 stoichiometry

Our pulldown data, SEC experiments, and XL-MS data all suggested that Mer2 might bind to mononucleosomes directly, perhaps providing additional affinity to the Spp1:nucleosome interaction. If true, this would be a previously unreported function of Mer2. We tested whether Mer2$^{FL}$ could bind to *unmodified* mononucleosomes in an SEC experiment, and surprisingly found that it formed a stable complex (*Figure 3A*). Given that Mer2 is a tetramer that binds two copies of Spp1, and given that Mer2 has been previously shown to form large assemblies on DNA (*Claeys Bouuaert et al., 2021*), we asked what the stoichiometry of a Mer2-mononucleosome complex was. To do this, we first used mass photometry (MP), a technique that determines molecular mass in solution at low concentrations based on the intensity of scattered light on a solid surface (*Young et al., 2018*). In MP, we observe a mix of three species, free Mer2 tetramer (measured at 127 kDa; theoretical mass 142 kDa), free mononucleosomes (measured at 187 kDa; theoretical mass 202 kDa), and a complex at 303 kDa, which most likely corresponds to a 4:1 complex of Mer2:nucleosomes (theoretical mass 344 kDa) (*Figure 3B*). The MP experiment was carried out at a protein concentration of 60 nM, which suggests that the dissociation constant ($K_D$) is somewhat less than 60 nM (at $K_D$ under equilibrium one would observe 50% complex formation, we observe less than 50% in *Figure 3B*). We next asked whether Mer2 might form larger assemblies on nucleosomes as higher concentrations. Using SEC-MALS (*Figure 3C*), we observed a complex of 341.0 kDa which matches a theoretical complex consisting of one Mer2 tetramer plus one mononucleosome (340 kDa, summarised in *Figure 3D*). Additionally, we observe a small fraction of a very high molecular weight assembly (though not an aggregate) of 10.96 MDa. This could be an oligomer of Mer2, of Mer2 on nucleosomes, or Mer2 on free DNA (it has recently been reported that Mer2 binds directly to DNA; *Claeys Bouuaert et al., 2021*). We also observe a shoulder on the Mer2-nucleosome peak, and find that this is a mixture of molecular masses (*Figure 3—figure supplement 1A*). Given that we observe Mer2 cross-links at the nucleosome DNA entry/exit site, we therefore tested whether Mer2 might simply be binding the free DNA ends on mononucleosomes. Using analytical EMSAs we found that Mer2 binds with a 6-fold higher apparent affinity to nucleosomes (5 nM vs. 30 nM) than to the same 167 bp DNA used to reconstitute the nucleosomes (*Figure 3E*). The discrepancy between $K_D$ determined by EMSA and the apparent $K_D$ from MP is presumably because EMSAs

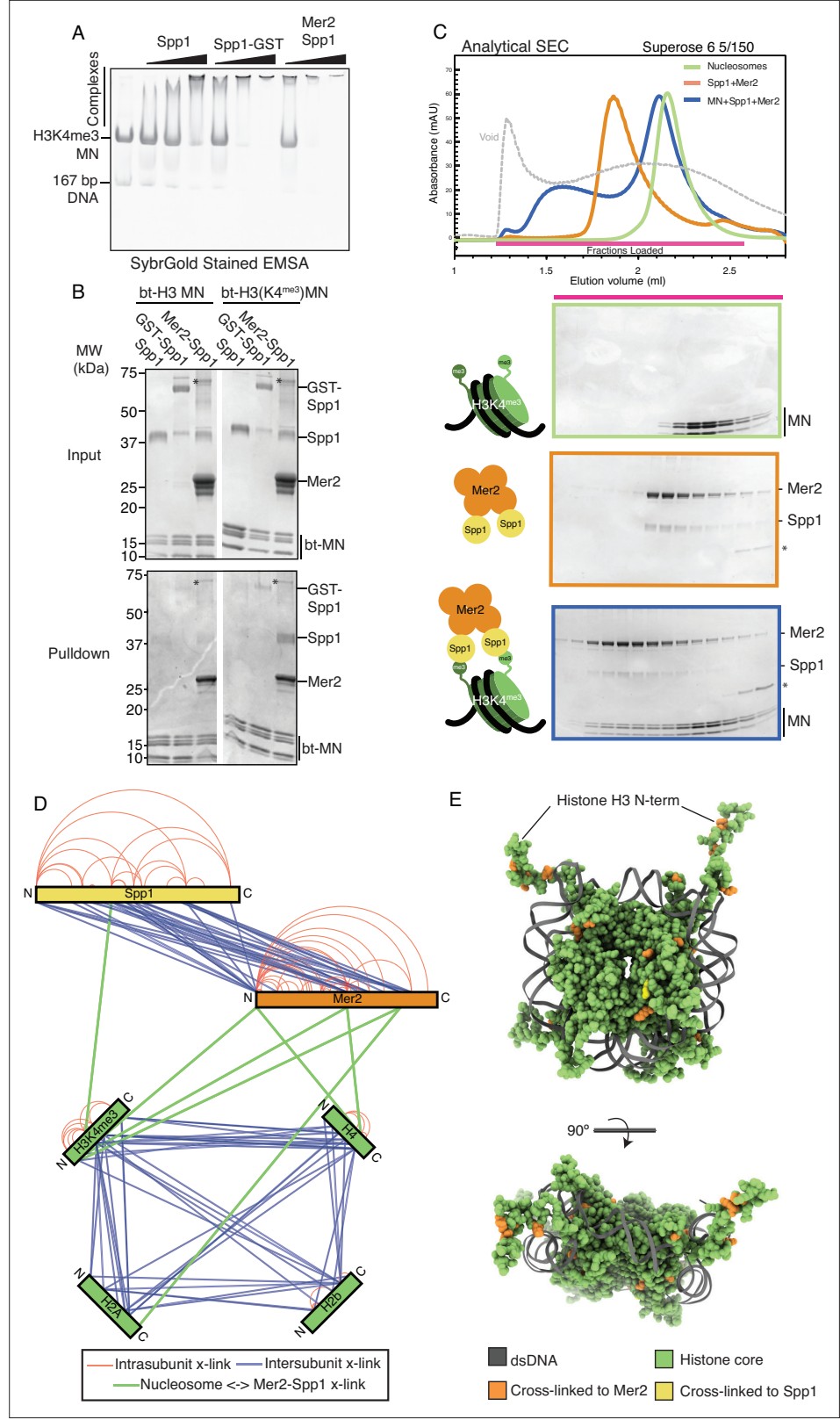

**Figure 2.** Mer2-Spp1 complex binding to H3K4$^{me3}$ mononucleosomes. (**A**) Electrophoretic mobility shift assay (EMSA) of different Spp1 variants on H3K4me3 nucleosomes. 0.2 μM of H3K4me3 nucleosomes were incubated with 0.33, 1, and 3 μM protein (Spp1, Spp1 with an N-terminal GST fusion, and Mer2-Spp1). Gel was post-stained with SYBRGold. (**B**) Biotinylated mononucleosome pulldown of different Spp1 variants. 0.5 μM nucleosomes

*Figure 2 continued on next page*

*Figure 2 continued*

wrapped with 167 bp of biotinylated DNA (either with or without the H3K4me3 modification) were incubated with 1.5 µM protein (same proteins as in A). Samples were taken for the input before incubation with streptavidin beads. The beads were then washed and eluted with 1× Laemmli buffer. Input and elution samples were run on a 10–20% SDS-PAGE gel and stained with InstantBlue. Asterisk marks residual uncleaved MBP-Mer2 in the Mer2-Spp1 lanes. (**C**) Size exclusion chromatography coupled to multi-angle light scattering (SEC) analysis of Mer2-Spp1-MN complex. 50 µL of 5 µM H3K4me3 mononucleosomes (green), 50 µL of 5 µM Mer2-Spp1 complex (orange), and 50 µL of a 1:1 (5 µM of each) mixture (blue) were run on a Superose 6 5/150 column. The same fractions were loaded in each case (magenta line) onto an SDS-PAGE gel and stained with InstantBlue. (**D**) Cross-linking coupled to mass spectrometry (XL-MS) analysis of the Mer2-Spp1-mononucleosome complex. Samples were cross-linked with disuccinimidyl dibutyric urea (DSBU), and data analysis carried out according to the Materials and methods. Cross-links were filtered so as to give a 1% false discovery rate. Figure was prepared using XVis (***Grimm et al., 2015***). Intramolecular cross-links are shown in red, intermolecular cross-links shown in blue. Cross-links between Mer2 and nucleosomes, and Spp1 and nucleosomes shown as thick green lines. (**E**) Model of nucleosome cross-links. The nucleosomes proximal cross-links from (D) were modelled onto a crystal structure of a *Xenopus laevis* nucleosome (PDB ID 1K × 5; ***Davey et al., 2002***). Those histone residues that cross-linked to Mer2 are coloured in orange, those that cross-link to Spp1 in yellow. A side and top-down view of the nucleosome are provided. DNA is coloured in dark grey, and histone residues that did not cross-link to Spp1 or Mer2 are coloured green.

The online version of this article includes the following source data for figure 2:

**Source data 1.** Raw gel data for electrophoretic mobility shift assays (EMSAs) (triplicate), pulldowns, and size exclusion chromatography (SEC) experiments.

are non-equilibrium experiments, carried out by necessity at very low salt (***Fried and Bromberg, 1997***).

Next we asked what effect using a smaller length of DNA to reconstitute nucleosomes might have. We reconstituted histone octamers on 147 bp DNA from now on referred to as nucleosome core particles (NCP; ***Lohr and Van Holde, 1975***; ***Sollner-Webb et al., 1976***). We found that with no free DNA ends Mer2 did bind with a lower affinity to NCPs, but nonetheless still with an apparent $K_D$ of ~40 nM (***Figure 3—figure supplement 2A***). We then asked whether mutating the common binding site, the 'acidic patch' on H2A (E56T-E61T-E64T-D90S-E91T-E92T) (***Kalashnikova et al., 2013***) might have an effect on Mer2 binding. In order to enhance potential differences in binding, this was also done on NCPs. Mer2 bound to NCP acidic patch mutants essentially as well as wildtype NCPs (***Figure 3—figure supplement 2A***). We then asked whether Mer2 might be recognising the histone tails. We therefore prepared 'tailless' NCPs (see Materials and methods). Surprisingly Mer2 bound very tightly to tailless NCPs with an apparent $K_D$ of ~5 nM (***Figure 3—figure supplement 2A***). We suggest that this might indicate that under the conditions of an EMSA, the histone tails are shielding either the histone cores or the NCP DNA and interfering with binding by Mer2.

We next asked which region of Mer2 might be involved in binding nucleosomes (reconstituted with 167 bp DNA). Initially we carried out EMSAs with Mer2$^{core}$, Mer2$^{\Delta N}$, and Mer2$^{\Delta C}$. Mer2$^{\Delta C}$ and Mer2$^{\Delta N}$ appeared to bind slightly weaker than Mer2$^{FL}$ ($K_D$ ~12.5 and ~30 nM, respectively), whereas Mer2$^{core}$ showed no binding at all (***Figure 3—figure supplement 2B***). Apparently there was equal contribution to nucleosome binding from both termini. In order to refine this further, we carried streptavidin pulldown using mononucleosomes reconstituted with biotinylated DNA against different Mer2 constructs. This approach had the advantage of being able to use a more physiological, and as such more stringent, buffer. We confirmed that the Mer2$^{core}$ did not bind nucleosomes, but neither did Mer2$^{\Delta C}$, strongly suggesting that the main nucleosome interaction region of Mer2 lies in the C-terminal 58 amino acids (***Figure 3F***), with some additional contribution from the N-terminus of Mer2 (summarised in ***Figure 3G***). We propose that Mer2, in addition to enabling the 'dimerisation' of Spp1 via its central tetramerisation domain, provides a direct binding interface with nucleosomes.

## Mer2 binding of Hop1 requires 'unlocking' the C-terminus of Hop1

In addition to Spp1, several lines of evidence point to an association between Mer2 and meiotic HORMA domain-containing factors. In fission yeast and mouse, the functional homologs of Mer2 (named Rec15 and IHO1, respectively) have been shown to interact with Hop1/HORMAD1 (***Kariyazono et al., 2019***; ***Stanzione et al., 2016***). Likewise, in budding yeast Mer2 exhibits a chromatin-association pattern which is very similar to Hop1 (***Panizza et al., 2011***). Meiotic HORMA proteins are integral members of

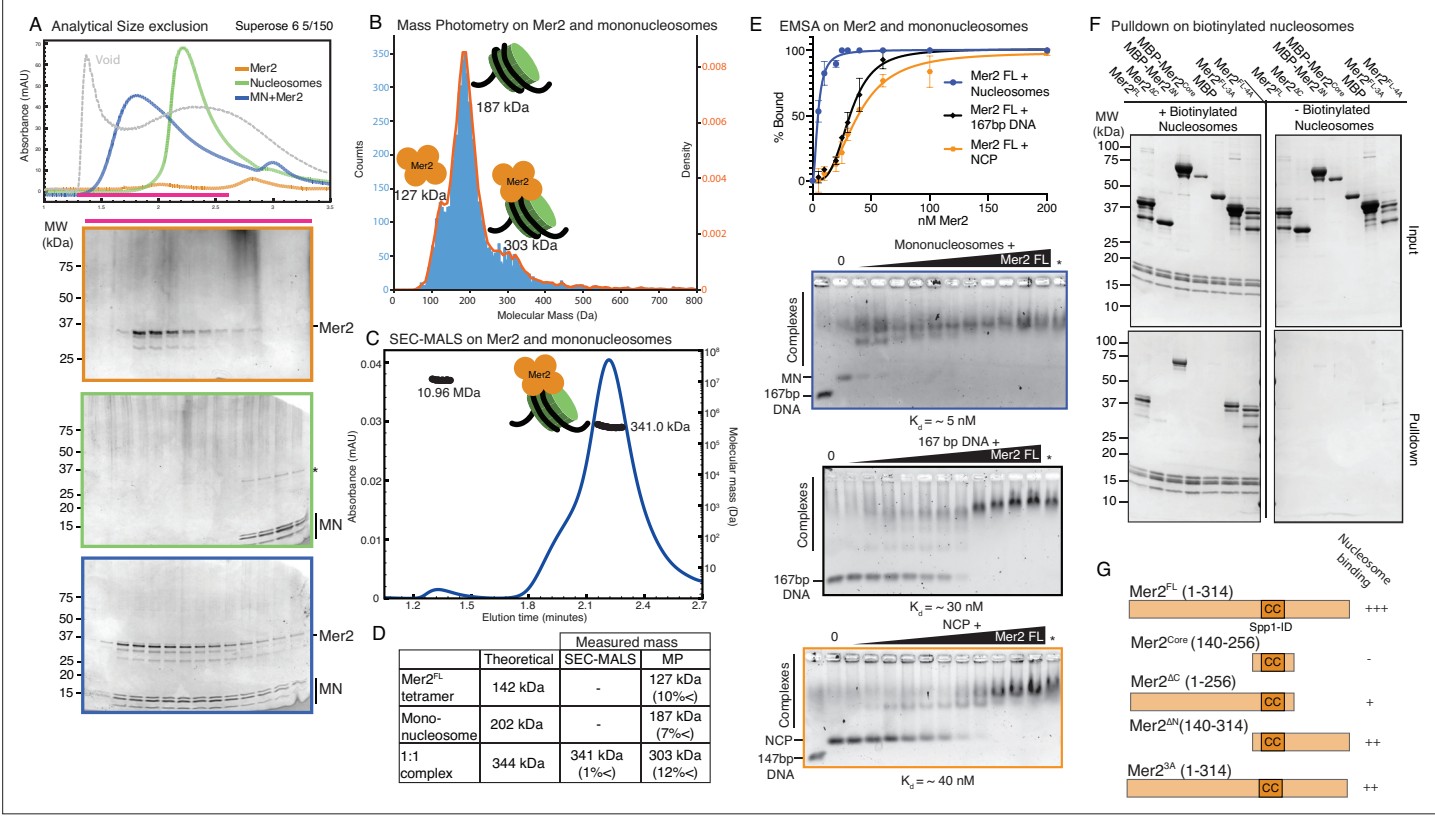

**Figure 3.** Mer2 binds directly to nucleosomes. (**A**) Size exclusion chromatography (SEC) analysis of Mer2-MN complex. 50 µL sample of 5 µM mononucleosomes (green), 50 µL of 5 µM full-length untagged Mer2 (orange), and a mixture of Mer2 and mononucleosomes (blue) were run on a Superose 6 5/150 column. The same fractions were loaded in each case (magenta line) onto an SDS-PAGE gel and stained with InstantBlue. (**B**) Mass photometry of Mer2 and mononucleosomes. 60 nM of Mer2 and mononucleosomes were mixed and analysed using a Refeyn One mass photometer. Three separate species were identified and the molecular mass determined using a molecular mass standard curve created under identical buffer conditions. Negative data points (i.e. unbinding events) were excluded. (**C**) SEC coupled to multi-angle light scattering (SEC-MALS) on Mer2-MN complex. Full-length, untagged Mer2 (Mer2^FL) was incubated with mononucleosomes and subject to SEC-MALS. Absorbance at 280 nm was constantly monitored (blue trace). Two distinct species were observed, one at 341 kDa, and another at 10.96 MDa (black trace). (**D**) Summary of molecular mass values for Mer2 FL, mononucleosomes (with 167 bp DNA), and a 1:1 complex. (**E**) Electrophoretic mobility shift assays (EMSAs) of untagged Mer2 FL. Mer2 was titrated against a constant 5 nM concentration of mononucleosomes (blue), 167 bp '601' DNA (black), or nucleosome core particle (orange). Binding curves were derived based on the Mer2-dependent depletion of free nucleosomes, DNA or NCP, and based on four independent experiments with error bars indicating the SD. Asterisk denotes the background SYBRGold staining from the highest protein concentration alone. The Hill coefficient was added to improve the goodness of fit (h = 1.9 for nucleosomes, 3.3 for DNA and 2.6 for NCP). (**F**) Pulldowns of Mer2 constructs with nucleosomes. Biotinylated nucleosomes (left panel) were incubated with different Mer2 constructs (as indicated), and samples taken for the input gel. The complexes were captured using streptavidin beads, washed, and eluted in 1× Laemmelli buffer for the pulldown gel. A control experiment (right panel) was conducted without biotinylated nucleosomes to measure non-specific Mer2 interaction with the streptavidin beads. (**G**) Domain cartoons of Mer2 summarising the different apparent affinities that each Mer2 construct has for mononucleosomes. Based on **E and F** and *Figure 3—figure supplement 2*.

The online version of this article includes the following source data and figure supplement(s) for figure 3:

**Source data 1.** Raw gel data for electrophoretic mobility shift assays (EMSAs) (triplicate), pulldowns, and size exclusion chromatography (SEC) experiments.

**Figure supplement 1.** Size exclusion chromatography coupled to multi-angle light scattering (SEC-MALS) on Mer2-nucleosome complexes.

**Figure supplement 2.** Electrophoretic mobility shift assays (EMSAs) on Mer2 binding to nucleosomes.

**Figure supplement 2—source data 1.** Raw gel data for EMSAs (triplicate).

the meiotic chromosome axis and are needed to recruit Mer2 to chromosomes (*Panizza et al., 2011*; *Stanzione et al., 2016*). Hop1 (like most other known HORMA domains) can exist in two topological states ('open/unbuckled' [O/U] or 'closed' [C]), in which the closed state can embrace a binding partner via a closed HORMA-closure motif 'safety belt' binding architecture (*West et al., 2018*). The

closure motif (also referred to as CM) is a loosely conserved peptide sequence encoded in HORMA binding partners. The meiotic HORMA proteins are unique among HORMA proteins in the fact that they contain a CM at the end of their own C-terminus (Hop1 residues 585–605; *West et al., 2018*), endowing these factors with the ability to form an intramolecular (closed) HORMA-CM configuration. The association of Hop1 with chromosomes is mediated by an interaction with Red1 which depends upon a similar CM:HORMA-based interaction. The CM of Red1 (located 340–362; *West et al., 2018*) binds to Hop1 with a higher affinity than the CM of Hop1 (*West et al., 2018*). There is mounting evidence in budding yeast and *Arabidopsis*, that, in addition to a chromosomal pool, a significant pool of Hop1 is non-chromosomal (in the nucleoplasm or cytoplasm) (*Herruzo et al., 2021*; *Raina and Vader, 2020*; *Yang et al., 2020*). Once bound to its own CM, Hop1 should not be able to interact with its chromosomal axis binding partner Red1. Due to the high local concentration of the intramolecular CM, free Hop1 is expected to rapidly transition from the 'open/unbucked' into the intramolecular 'closed' state (*West et al., 2018*). The closed state (whether it is intramolecular or, for example, with the CM of Red1) can be reversed by the action of the AAA+ ATPase Pch2/TRIP13 (*Rosenberg and Corbett, 2015*; *Vader, 2015*). As such, within the nucleoplasm/cytoplasm, Pch2/TRIP13 activity serves to generate enough 'open/unbuckled' Hop1 that is proficient for incorporation into the chromosomal axis, via a CM-based interaction with Red1. On the other hand, when recruited to chromosomes, that same Pch2/TRIP13 activity is expected to dismantle Hop1-Red1 assemblies, as such leading to removal of Hop1 from chromosomes (*Deshong et al., 2014*; *Subramanian et al., 2016*).

We tested the ability of Mer2 to interact with the proteins of the meiotic axis (i.e. Hop1 and Red1), with a focus on Hop1. Initially, we purified Hop1 using an N-terminal 2x-Strep-II tag and used it to pull down Mer2. We observed a faint band corresponding to Mer2 in the Hop1 pulldown, indicative of a weak interaction (*Figure 4—figure supplement 1*, lane 1, control in lane 2). Note that based on the high relative concentration of the CM in Hop1, this Hop1 is expected to largely consist of (intramolecular) closed Hop1. We then co-expressed Red1-MBP containing a I743R (from hereon referred to as Red1$^{I743R}$-MBP) mutation with Hop1 in insect cells. The I743R mutation should prevent Red1 from forming filaments, but still allow Red1 to form tetramers (*West et al., 2019*). Given that we have an excess of Hop1 in our Hop1-Red1 purification, we carried out a pulldown on the MBP tag of Red1 (using amylose beads) with Mer2 as prey. In this case, we observed considerably more Mer2 binding, when measured relative to Hop1, its putative direct binding partner (*Figure 4—figure supplement 1*, lane 3, control in lane 4). We quantitated the Mer2 intensity relative to the Hop1 band in both pull-downs, and from three independent experiments, and observed an ~6-fold increase in Mer2 binding (*Figure 4A*). We reasoned that this difference could either be due to Red1 interacting directly with Mer2, or that Red1 induces a conformational change in Hop1 that facilitates Mer2 binding. To test the former idea we purified Red1$^{I743R}$-MBP in the presence or absence of Strep-Hop1. Using amylose affinity beads to capture the MBP moiety of Red1 we tested the capture of Mer2 both in the presence and in the absence of Hop1 (*Figure 4B*). We detected no Mer2 interaction when pulling on Red1$^{I743R}$-MBP in the absence of Hop1. As such, we conclude that Mer2 does not have significant affinity for Red1. These data argue that Hop1, when bound to Red1, is acting as an efficient recruiter of Mer2 to this complex. How could this increased affinity of Hop1 for Mer2 be influenced by association with Red1? Based on the known biochemical basis of the Hop1-Red1 interaction, we can imagine that the interaction of Red1 with Hop1 'releases' the C-terminus of Hop1 which could create a 'chain' of Hop1 moieties (akin to what has been observed in *Caenorhabditis elegans*; *Kim et al., 2014*) or could simply liberate the C-terminus of Hop1 for binding to non-self partners. We propose that such configurations would create or expose Mer2-specific binding interfaces (which can conceivably be located on either the HORMA domain or within the C-terminal non-HORMA domain) that are otherwise shielded when Hop1 is bound intramolecularly to its own C-terminal CM. This predicts that impairing intramolecular CM binding to the HORMA domain should 'unlock' the binding ability of Hop1 with Mer2, regardless of Red1 presence. To test this prediction and to delineate the binding interface between Mer2-Hop1, we created two additional Hop1 constructs, one where the N-terminal HORMA domain is missing (Hop1$^{\Delta HORMA}$) and another where the conserved lysine in the CM has been mutated to alanine (Hop1$^{K593A}$) disrupting the ability of the CM of Hop1 to interact with its HORMA domain thus forcing Hop1$^{K593A}$ into an 'unlocked' state (*West et al., 2018*). Both Hop1$^{\Delta HORMA}$ and Hop1$^{K593A}$ were purified with an N-terminal 2xStrep-II tag (as for Hop1$^{WT}$) and their ability to bind to Mer2 was tested. In order to prevent background binding, stringent binding conditions were used. We

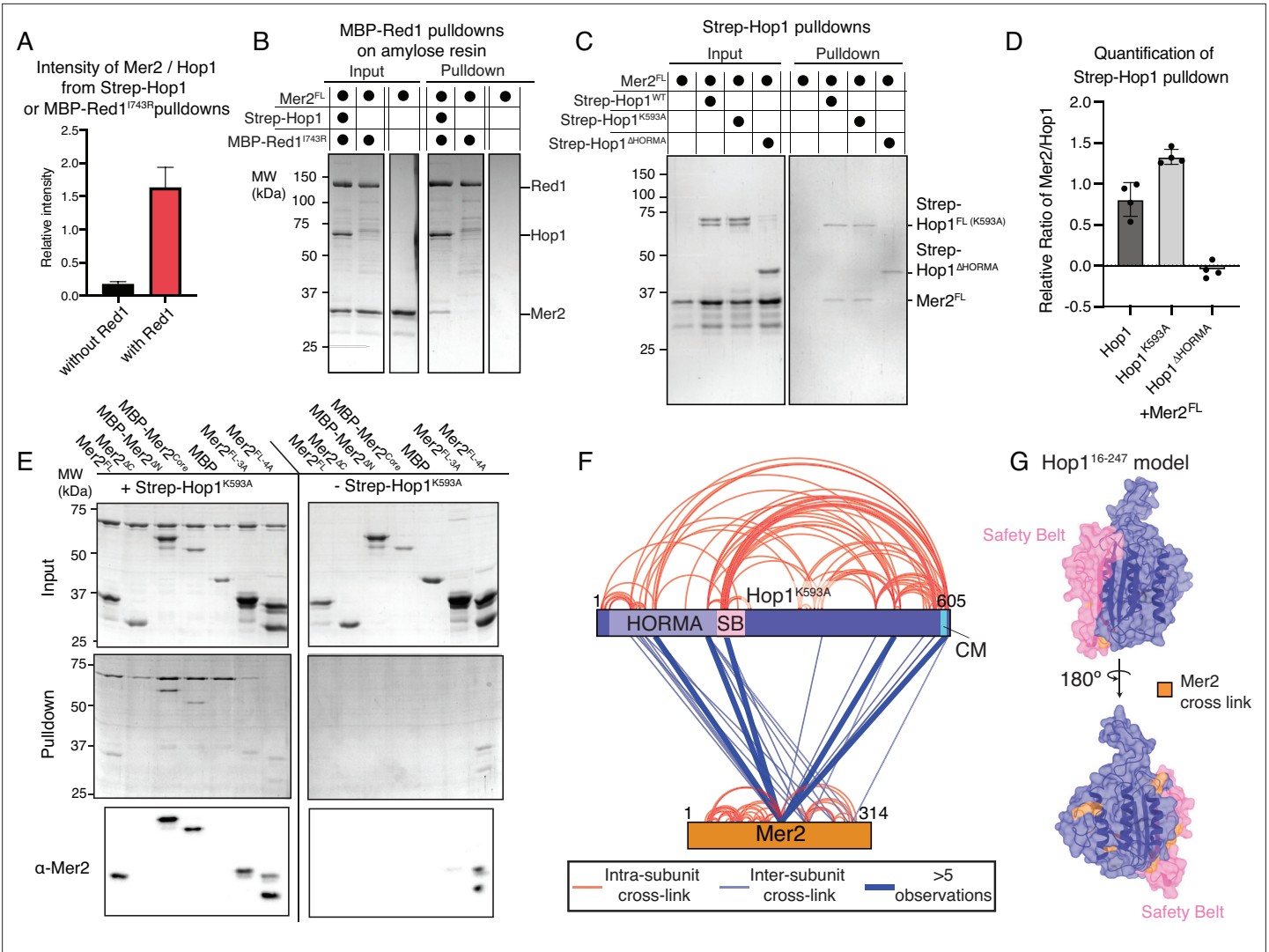

**Figure 4.** Mer2 binds to the axis via Hop1. (**A**) Quantification of Streptactin pulldowns, or amylose pulldowns on either Strep-Hop1 (black) or the Strep-Hop1/Red1[I743R]-MBP (red) complex against Mer2 (from gels shown in ***Figure 4—figure supplement 1***). In all cases the intensity of Mer2 was quantified as a factor of Hop1 intensity. Error bars are based on the SD from three independent experiments. (**B**) Amylose pulldown of the co-expressed Red1[I743R]-MBP/Hop1 complex and Red1[I743R]-MBP alone against Mer2. Pulldown was carried out as described in Materials and methods. In the rightmost lane Mer2 alone was used as a negative control on the beads. (**C**) Streptactin pulldown of different 2xStrepII-Hop1 constructs (as indicated) against untagged Mer2[FL]. N-terminally 2xStrep-II tagged Hop1 constructs were purified as indicated and used as a bait against Mer2 full length. Pulldown was carried out as described in the Materials and methods. (**D**) Quantification of experiments equivalent to those shown in **C**. Intensity of Mer2 was quantified as a factor of Strep-Hop1 intensity. Error bars are based on the SD from four independent experiments. (**E**) Streptactin pulldown of Strep-Hop1[K593A] as bait to capture different Mer2 constructs as indicated. The rightmost pulldown is the control showing the non-specific binding of different Mer2 constructs to the Streptactin beads. An anti-Mer2 Western was carried out (lower panel) to confirm the specific interactions due to the weakness of the Mer2 bands in a coomassie stained gel. (**F**) Representation of cross-linking mass-spectrometry (XL-MS) analysis of the Hop1[K593A]-Mer2 complex. Cross-links were filtered so as to give a <1% false discovery rate. Figure was prepared using XVis (***Grimm et al., 2015***). Intramolecular cross-links are shown in red, intermolecular cross-links shown in blue. Thick blue lines represent those intermolecular cross-links that were observed five or more times. The HORMA domain of Hop1, the safety belt (SB), and the closure motif (CM) are highlighted. (**G**) Homology model of the Hop1 HORMA domain (residues 16–247) using open-Mad2 as a template (PDB 1duj; ***Luo et al., 2000***) is shown in two orientations. The safety belt is highlighted in pink. Those residues that cross-link to Mer2 are highlighted in orange.

The online version of this article includes the following source data and figure supplement(s) for figure 4:

**Source data 1.** Raw gel data for pulldowns (triplicate and singular), anti-Mer2 Western blot raw blot data.

**Figure supplement 1.** Mer2 pulldowns on Hop1-Red1 complexes.

**Figure supplement 1—source data 1.** Raw gel data for pulldowns.

saw that both Hop1$^{WT}$ and Hop1$^{K593A}$ bind to Mer2$^{FL}$, whereas Hop1$^{\Delta HORMA}$ does not (**Figure 4C**). Quantification of the pulldown analyses showed that Hop1$^{K593A}$ has ~2-fold stronger binding to Mer2 than does Hop1$^{WT}$ (**Figure 4D**). Based on these data, and taking into consideration the effect of Red1 on the Hop1-Mer2 interaction, we suggest the following model: Mer2 binding to Hop1 is driven largely by HORMA domain-mediated interactions, and promoted under conditions where the HORMA is not bound to its intramolecular CM. The most parsimonious molecular explanation of this behavior is that the C-terminal region of Hop1, when kept in close proximity of the HORMA domain (by virtue of CM-HORMA binding), can sterically interfere with the establishment of a Mer2-Hop1 association. Furthermore, the robust Mer2 binding we observe for the Hop1-Red1 complex (**Figure 4B**) suggests that there is additional contribution from Red1, likely through promoting the closed conformation of the HORMA domain through Red1-CM-HORMA domain binding.

We next asked which region of Mer2 is necessary for Hop1 interaction. Using N-terminally 2xStrepII tagged Hop1$^{K593A}$ as bait, we queried a variety of Mer2 constructs using Streptactin beads (**Figure 4E**). Due to the weak staining of Mer2 under these conditions, we also carried out an anti-Mer2 Western blot (**Figure 4E** lower panel). We determined that Hop1 was capable of interacting, apparently equally well, with all Mer2 constructs, except for Mer2$^{\Delta C}$. Why would the Mer2$^{core}$ be capable of binding to Hop1, but the ΔC not? The answer may lie in the complex arrangement of the Mer2 coiled-coils and thus N- and C-termini of Mer2 relative to one another and to the core of Mer2. As such one could imagine that part of the interaction region for Hop1 lies in the core of Mer2, which might be shielded by the N-terminus when the C-terminus of Mer2 is not present. These results hint at a complex regulation, on the Mer2 side, underlying Mer2-Hop1 interaction, and we believe that high-resolution structural studies of the Mer2-Hop1 interface should eventually be able to provide deeper insight into this interaction. Taken together we propose a new model for Mer2 recruitment to the meiotic axis in which a 'locked' Hop1 cannot bind to Mer2 but once incorporated into Red1, Hop1 recruits Mer2 to the meiotic axis.

We sought to further characterise the interaction between Mer2 and Hop1 by again making use of XL-MS and the DSBU cross-linker (**Figure 4F**). We observed the most cross-links between the core of Mer2 (consistent with the binding of Mer2$^{core}$ to Hop1, **Figure 4E**) and both the HORMA domain of Hop1 and the C-terminal part of the protein (few cross-links were detected within the putative PHD/wHTH like region of Hop1, located in the non-HORMA domain C-terminal part of Hop1; **Ur and Corbett, 2021**). We also observed additional cross-links with the C-terminal half of Mer2, again consistent with the pulldown data (**Figure 4E**). Using a model generated using open-Mad2 as a template (**Luo et al., 2000**), we mapped Mer2-Hop1 cross-links onto the HORMA/SB domain of Hop1 (**Figure 4G**). This revealed that multiple Mer2 cross-links were concentrated on one 'face' of the HORMA domain, whereas the other 'face' of the HORMA domain is essentially free of Mer2-Hop1 cross-links. We observed several cross-links in close proximity to the safety belt of the HORMA domain of Hop1 (**Figure 4G**). While we assume that the Hop1$^{K593A}$ used in the cross-linking is mostly 'unlocked', we acknowledge that the safety belt likely has a somewhat different orientation in 'unlocked' Hop1 vs. 'open' Mad2 (**West et al., 2018**). Taken together these data reveal additional molecular details of the Mer2-Hop1 binding mode. The concentration of Mer2 cross-links on one face of Hop1 hints at a potential role for Red1 in enforcing a particular Hop1 orientation that is compatible with efficient recruitment of Mer2.

## A conserved N-terminal motif in Mer2 is essential for meiosis

Due to the potential difficulties in assigning defined interaction regions within Mer2 when using truncation constructs, as so far described, we instead aimed to obtain separation of function alleles by introducing selected point mutants. Making use of sequence alignments from (evolutionary closely and more distantly related) Mer2 orthologs, we identified a conserved region, in the N-terminal domain between amino acids 52 and 71 (**Figure 5A**, **Figure 5—figure supplement 1**), that has also been previously annotated as Mer2 SSM1 (**Tessé et al., 2017**). This particular stretch of amino acids stands out in the protein sequence of Mer2 (and homologs) since, in addition to the central coiled-coil region, this region is one of the few regions which shows sequence similarities across evolutionary distant species (such as yeast and human). To probe a potential function of this conserved region, we created two different alleles containing mutations in this region, which we here refer to as *mer2-3a* and *mer2-4a*. In *mer2-3a*, we mutated three conserved residues *W58*, *K61*, and *L64* to alanine (**Figure 5A**,

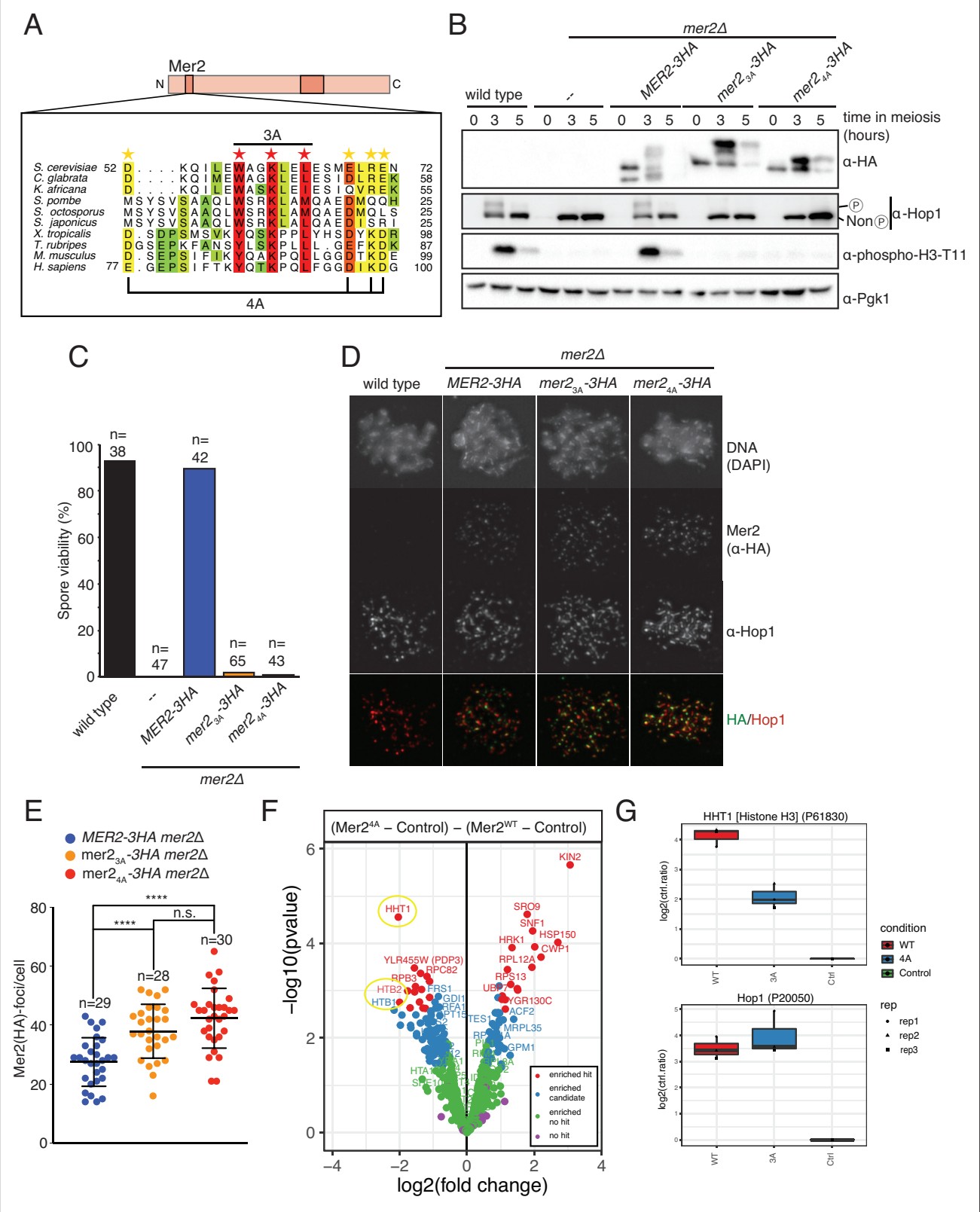

**Figure 5.** Mer2 3A and 4A mutations lead to inviable spores. (**A**) Sequence alignments of Mer2 orthologs from multiple clades revealed a previously undescribed conservation in the N-terminal region. We created two mutants, the '3A' mutant (W58A, K61A, L64A; red stars) based on three universally conserved residues and the '4A' mutant (D52A, E68A, R70A, E71A; yellow stars). The periodicity of the red star conserved residues, combined with some secondary structure predictions, suggest that these residues might be on the same face of an alpha-helix. (**B**) Western blot analysis of meiotic

*Figure 5 continued on next page*

*Figure 5 continued*

yeast cultures of wildtype, *mer2Δ, MER2-3HA mer2Δ, mer2-3a-3HA mer2Δ,* and *mer2-4a-3HA mer2Δ* strains. Time of induction into the meiotic program indicated. See Yeast strains used in this study for strain information. Pgk1 was used as a loading control. Please note the indicated phospho-Hop1, anti-phospho histone H3 was used to measure the activation of Mek1 kinase in response to DSB formation. (**C**) Quantification of spore viabilities of wildtype, *mer2Δ, MER2-3HA mer2Δ, mer2-3a-3HA mer2Δ,* and *mer2-4a-3HA mer2Δ* strains. Number of dissected tetrads is indicated. See Yeast strains used in this study for strain information. (**D**) Representative images of meiotic chromosome spreads stained for Mer2 (α-HA [green], Hop1, and DNA [blue]) using 4′,6-diamidine-2′-phenylindole dihydrochloride (DAPI) from wildtype, *MER2-3HA mer2Δ, mer2-3a-3HA mer2Δ,* and *mer2-4a-3HA mer2Δ* strains at t = 3 hr after induction into the meiotic time course. (**E**) Quantification of the number of Mer2 foci on chromosome spreads of strains used in (**D**). Mean and standard deviation are indicated. *p ≤ 0.05, **p ≤ 0.01, and ****p ≤ 0.0001; n.s. (non-significant) >0.05. Mann-Whitney U test. Number of analysed cells is indicated. (**F**) Volcano plots of quantitative immunoprecipitation (IP) mass spectrometry of Mer2$^{WT}$ vs. Mer2$^{4A}$. Proteins are grouped into hits (red) and candidates (blue), according to the fold change and p-value. Proteins appearing on the left show reduced association with Mer2$^{3A}$ when compared with Mer2$^{WT}$, whereas proteins on the right show an increased association with Mer2$^{4A}$ when compared with Mer2$^{WT}$. Circled in yellow are the histone proteins Hht1 (histone H3) and Htb1/Htb2 (histone H2B). (**G**) Detail of Mer2$^{4A}$ vs. Mer2$^{WT}$ in IP quantitative mass spectrometry analysis for Hht1 (histone H3); top panel, and Hop1; lower panel. Association with Hht1 is reduced ~2-fold in Mer2$^{4A}$ whereas it is essentially unchanged for Hop1.

The online version of this article includes the following source data and figure supplement(s) for figure 5:

**Source data 1.** Raw Western blot data plus explanatory data.

**Figure supplement 1.** Mer2 sequence alignment.

**Figure supplement 2.** Immunoprecipitation mass spectrometry (IP-MS) of Mer2$^{WT}$ vs. Mer2$^{4A}$ (nucleosomes, Hop1, Red1).

red stars). The significance here is both in the high level of conservation, and, in the case of W58, in the presence of a bulky hydrophobic residue that might be involved in protein-protein interactions. Finally, the periodicity of the conserved residues combined with the secondary structure prediction hinted that these residues might be on the same face of an alpha-helical element (see *Figure 5—figure supplement 1*). *mer2-4a* contains the following mutations within the same domain: *D52A, E68A, R70A,* and *E71A* (*Figure 4A*, yellow stars). We note that the residues in *mer2-4a* are less well conserved than those in the *mer2-3a* allele, but we wondered whether they might function together or in separate pathways. We integrated plasmids carrying C-terminally 3HA-tagged versions of wildtype *MER2, mer2-3a,* or *mer2-4a* (note that all these constructs are driven by *pMER2*) in *mer2Δ* strains. All three constructs lead to comparable expression levels of Mer2 during meiotic prophase, although we note different mobility of the Mer2$^{3A}$-3HA or Mer2$^{4A}$-3HA as compared to Mer2-3HA, which might indicate altered post-translational modifications, such as phosphorylation or SUMOylation, since Mer2 was recently shown to be heavily SUMOylated including sites in the region covered by the 3A and 4A mutations (Mer2 K53 and K61; *Bhagwat et al., 2021*; *Figure 5B*). Alternatively, it might reflect an inherent effect of the introduced mutations on electrophoretic behavior (compare for example also the migrating patterns of wildtype Mer2 with Mer2$^{3A}$ on SDS-PAGE from our in vitro preparations, where meiosis-specific post-translational modifications are most likely absent; see *Figure 6B*).

The role of Mer2 in meiotic DSB activity is key in enabling faithful meiotic chromosome segregation and viable spore formation. Consequently, *mer2Δ* strains exhibit very strong spore viability defects, as reported previously (*Rockmill et al., 1995 Figure 5C*). As a first test of functionality, we investigated the effect of our *MER2* alleles on spore viability. Expression of Mer2-3HA in *mer2Δ* overcame the spore viability defect seen in cells lacking Mer2 (*Figure 5C*, blue bar) demonstrating functionality of our *MER2* expressing constructs. Strikingly, expression of Mer2$^{3A}$-3HA or Mer2$^{4A}$-3HA failed to restore spore viability in *mer2Δ*; these strains essentially behaved indistinguishably from the *mer2Δ* strain. This suggests that the designed mutants disrupt a functionality of Mer2 that is key to its role during meiotic prophase. When we probed for the activation of Mek1 kinase, which is regulated by DSB-dependent Mec1/Tel1 activation (*Chuang et al., 2012*; *Niu et al., 2005*), we noticed that expression of the Mer2$^{3A/4A}$ mutants (in contrast to wildtype Mer2) prevented the appearance of the phosphorylated version of histone H3-threonine 11, a bona-fide Mek1 substrate (*Kniewel et al., 2017*; *Figure 5B*). Together with the fact that expression of Mer2$^{3A/4A}$ mutants did not support the phosphorylation of Hop1 (mediated by Mec1/Tel1 as a response to DSBs) (*Carballo et al., 2008*; *Figure 5B*; as seen by an apparent electrophoretic shift in Hop1), these data show that the expression of our Mer2 mutants leads to defective Mec1/Tel1-Mek1 signaling. This hints that DSB formation might be negatively affected by the mutations that we introduced in Mer2.

Since the investigated region of Mer2 is located within the N-terminal region of the protein, we expected that these mutants would not disrupt the interactions with Spp1, Hop1, and nucleosomes

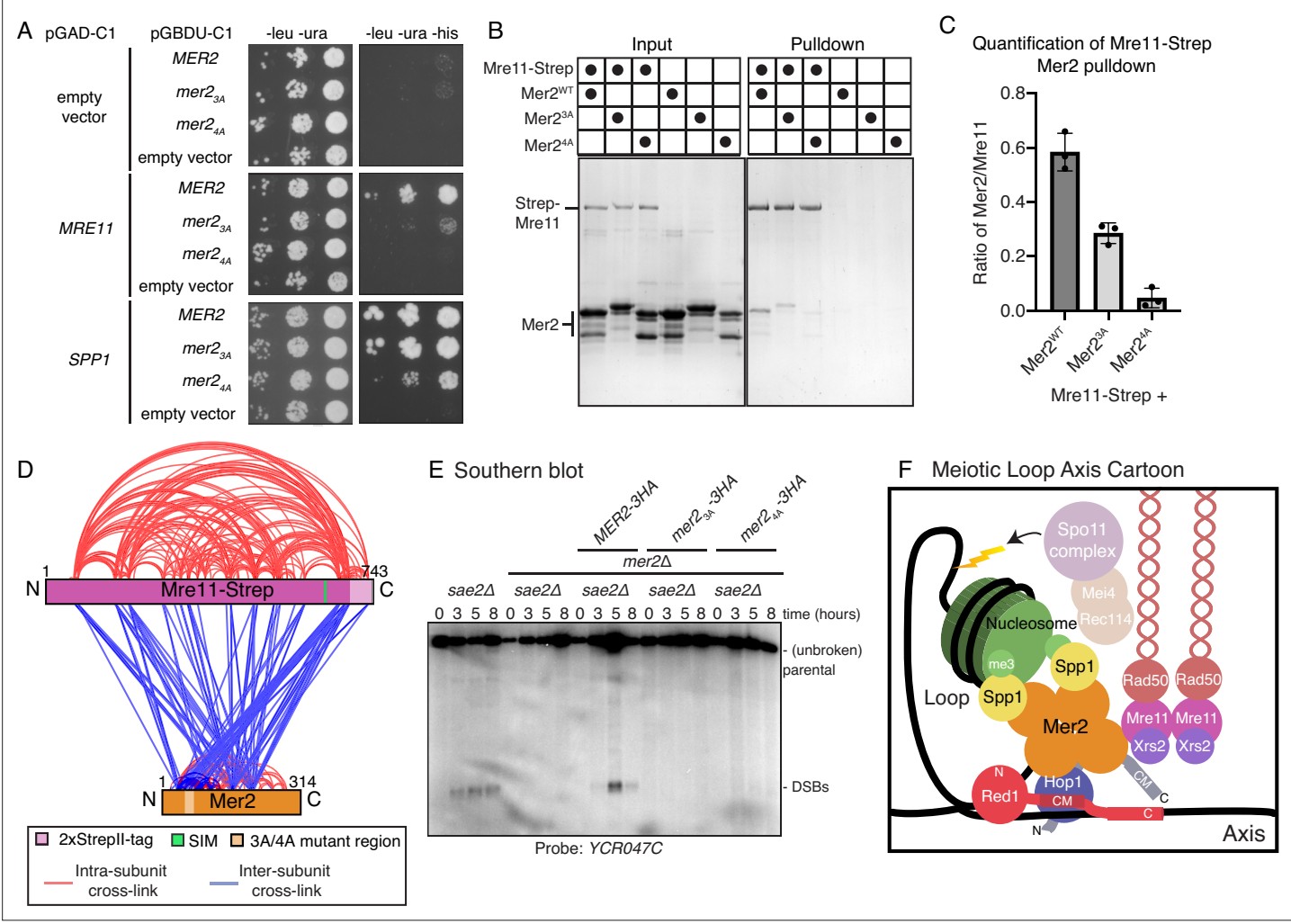

**Figure 6.** Mer2 binds directly to Mre11. (**A**) Yeast-2-hybrid experiments with Mer2[WT], Mer2[3A], and Mer2[4A] mutants. Yeast were transformed with the pGAD-C1 (activation domain) and pGBDU (DNA binding domain) plasmids as indicated. Cells were grown and pipetted onto non-selective (left) or selective plates (right) at three serial concentrations. (**B**) In vitro pulldowns between Strep-Mre11 and Mer2. C-terminally 2xStrepII-tagged Mre11 was used to capture Mer2[WT], Mer2[3A], and Mer2[4A] on Streptactin beads. Simultaneously Mer2[WT], Mer2[3A], and Mer2[4A] in the absence of Mre11-Strep were used to determine non-specific binding to the beads. (**C**) Quantification of Strep-Mre11 pulldowns against Mer2[WT], Mer2[3A], and Mer2[4A]. Intensity of Mer2 was quantified as a factor of Strep-Mre11 intensity. Error bars are based on the SD from three independent experiments. (**D**) Representation of cross-linking mass spectrometry (XL-MS) analysis of the Mre11-Mer2 complex. Cross-links were filtered so as to give a <1% false discovery rate. Figure was prepared using XVis (*Grimm et al., 2015*). Intramolecular cross-links are shown in red, intermolecular cross-links shown in blue. The location of the SIM is the highest scoring motif (score = 64.397) in Mre11 as determined by the GPS-SUMO prediction tool (*Zhao et al., 2014*). (**E**) Southern blot analysis comparing double-strand break (DSB) levels in *sae2Δ, mer2Δsae2Δ, MER2-3HA mer2Δsae2Δ, mer2-3a-3HA mer2Δsae2Δ,* and *mer2-4a-3HA mer2Δsae2Δ* strains. Cells were harvested at indicated time points post induction of meiosis and faster migrating DNA species (indicative of DSBs) were detected using a probe for the *YCR047C* locus. *sae2Δ* cells are resection-deficient and cannot repair DSBs. (**F**) Final meiotic loop axis cartoon. Summarises the novel interactions shown in the previous figures while highlighting the integration of Mre11 via its direct binding to Mer2, into the loop axis model. Briefly, a tetrameric Mer2 binds to two copies of Spp1, while also binding to nucleosomes. Mer2 is also able to interact directly with Hop1, in a manner that is enhanced when Hop1 is bound to Red1. Finally Mer2 also interacts with Mre11 via a conserved N-terminal motif.

The online version of this article includes the following source data and figure supplement(s) for figure 6:

**Source data 1.** Raw gel data for pulldowns.

**Source data 2.** Raw gel data for Southern blots.

**Figure supplement 1.** Yeast-2-hybrid (Y2H) with Mer2 and Spo11 complex components.

**Figure supplement 2.** Immunoprecipitations (IPs) of Mer2[WT], Mer2[3A], and Mer2[4A] against Rec114 and Mre11.

**Figure supplement 2—source data 1.** Raw gel data for western blots, plus explanatory data.

**Figure supplement 3.** IFs of Mer2[WT], Mer2[3A], and Mer2[4A] against Rec114 and Mre11.

**Figure supplement 4.** Summary of Mer2 interactions and organisation.

described above. In line with this idea, we observed that in vitro, recombinant Mer2[3A] and Mer2[4A] are able to interact with nucleosomes (*Figure 3F*) and Hop1 (*Figure 4E*), albeit with slightly lower affinity for nucleosomes as determined by EMSA (*Figure 3—figure supplement 2C*).

To attempt to trace the defect of these *MER2* alleles, we first investigated the association of Mer2 on spread meiotic chromosomes during meiotic prophase. We found that the recruitment of Mer2 to defined chromosomal foci was not disrupted in Mer2[3A] and Mer2[4A] expressing situations (*Figure 5D*). In contrast, quantification of the number of Mer2 foci that were observed during meiotic prophase revealed an increase in cells expressing these mutant proteins (*Figure 5E*). We currently do not understand the reason underlying this apparent increase in chromosome-associated Mer2 foci, but it could conceivably be related to known feedback regulation that ensures sustained DSB activity in conditions where chromosome synapsis fails or is delayed (*Thacker et al., 2014*). In any case, recruitment of Mer2 to chromatin-associated foci was not disrupted, suggesting that Mer2[3A] and Mer2[4A] are proficient to interact with upstream factors that lead to chromatin recruitment.

In order to identify a potential protein-protein interaction that is disrupted in the Mer2 mutants, we carried out a quantitative mass spectrometry analysis of samples immunoprecipitated from meiotic *mer2Δ* SK1 yeast expressing Mer2[4A]-3HA or Mer2[WT]-3HA. Wildtype (carrying an untagged *MER2* allele) SK1 cells were used as a control. In order to prevent possible confounding effects due to potential differences in DSB forming potential (based on the effect on spore viability and the decrease in phospho-histone H3-T11 signal drop we observed in our mutants, *Figure 5B and C*), we performed the analysis in a DSB-deficient condition (by using the catalytic-dead *spo11-Y135F* allele) in both Mer2[4A]-3HA and Mer2[WT]-3HA. The summary of the differences between Mer2[WT] and Mer2[4A] are shown in the volcano plot in *Figure 5F*. In summary, the biggest change we see in the Mer2[4A] mutant vs. Mer2[WT] is a reduced binding to Hht1 (histone H3). We also saw a reduced binding to the other histones Htb1 and Htb2 (histone H2B). Interaction with Hop1 was largely unaffected (*Figure 5G*) – in agreement with our in vitro analysis (*Figure 4E*), as was association with Spp1 and Red1; the latter presumably associated via association with Hop1 (*Figure 5—figure supplement 2A and B*). Taken together, it appears from these analyses that Mer2[4A] exhibits reduced direct nucleosome association. Nonetheless, the effect is likely mild, since we see in vitro association is at most reduced 5-fold in EMSAs for Mer2[3A] (*Figure 3—figure supplement 2C*), and both Mer2[3A] and Mer2[4A] robustly interact with nucleosomes in a pulldown (*Figure 3F*), and Mer2 still localises to chromatin (*Figure 5D and E*).

What might be the cause of the penetrant phenotype seen in Mer2[3A] and Mer2[4A]? Since HORMA proteins are suggested to drive recruitment of Mer2 to the chromosome axis (*Kariyazono et al., 2019*; *Panizza et al., 2011*; *Stanzione et al., 2016*; and see our earlier observations), these results argue that the defect that is triggered by *mer2-3a* and *mer2-4a* is independent of its interaction with Hop1. Furthermore, the observation that spore viability was severely impaired in cells expressing *mer2-3a* and *mer2-4a* is in contrast with the relatively mild phenotype caused by the specific disruption of the Mer2-Spp1 interaction (*Adam et al., 2018*). Finally, our Western blots analysis revealed a loss of Mek1-dependent histone H3-pT11 phosphorylation (a proxy for meiotic DSB formation) in the Mer2[3A/4A] mutants (*Figure 5B*, and see above). Together, these observations, in combination with the strong evolutionary conservation of the affected region, argue that the observed defects are caused by the disruption of yet another key Mer2 interaction, potentially with another essential DSB factor, leading to possible defects in DSB formation.

## The Mer2[3A] and Mer2[4A] mutants are defective in Mre11 binding and DSB formation

Strikingly, we detected no Spo11-associated machinery in the Mer2[WT] IP (*Figure 5—figure supplement 2C* and Supplementary mass spectrometry data), despite a large total number of enriched proteins. We therefore postulated that we might be missing key Mer2 interactions – which are potentially difficult to detect using in vivo biochemical purification (ostensibly due to transient/low affinity interactions). The disruption of those interactions, as a result of the Mer2 mutants, might explain observed in vivo phenotypes. As such, we carried out a directed Y2H analysis against previously identified Mer2 interactors involved in DSB formation (Xrs2, Mre11, Ski8, Spo11, Rec104) (*Arora et al., 2004*) and Spp1 (as a positive control). Analysis of some of these interactions was complicated by the presence of apparent self-activation in our Y2H system. Nonetheless, we observed a clear difference in binding between the Mer2 and Mre11 when comparing *MER2* to the *mer2-3a* or *mer2-4a* mutants

(*Figure 6A*, *Figure 6—figure supplement 1*). Importantly, these mutants were proficient to interact with Spp1, similar to what was indicated by our in vitro binding assays and in accordance with the mass spectrometry data (*Figure 5—figure supplement 2A*).

We aimed to confirm the interaction between Mer2 and Mre11 by other means, and sought to query whether the interaction with other DSB factors was not perturbed in Mer2$^{3A/4A}$ mutants. We initially used in vivo co-immunoprecipitation (co-IP) to test for Mer2 interaction with Mre11 and Rec114 (*Figure 6—figure supplement 2A and B*). Using this setup, we were not able to detect Mre11 nor Rec114, although we reliably detected Hop1 in Mer2$^{WT}$, Mer2$^{3A}$, and Mer2$^{4A}$ co-IPs, indicating efficient purification of a relevant assembly. The apparent discrepancy between our Y2H and co-IP experiments might reflect an inherently transient or infrequently occurring interaction between Mer2 and Mre11 (and Mer2 and Rec114), precluding detection using our current in vivo methodology. We note that the results obtained with our co-IP experiments (i.e. no detected interaction with Mer2 or Rec114, but robust interaction with Hop1) were similar to the results we obtained using our IP-MS approach comparing Mer2$^{WT}$ and Mer2$^{4A}$ (*Figure 5F and G*). Therefore, we sought to corroborate an interaction between Mer2 and Mre11 using purified components. To this end, we expressed and purified recombinant Mre11 carrying a C-terminal 2xStrep-II tag using baculovirus-based expression. We then used this as a bait to pull down Mer2$^{WT}$, Mer2$^{3A}$ and Mer2$^{4A}$. We detected a robust interaction between Mer2$^{WT}$ and Mre11, and in line with our earlier result, we observed a reduced interaction between Mre11 and Mer2 when Mer2$^{3A}$ and Mer2$^{4A}$ were used in the pulldown assay (*Figure 6B*). We quantified the amount of Mer2 in the pulldown as a factor of Mre11 (*Figure 6C*): Mer2$^{3A}$ leads to a 2-fold reduction in binding, and Mer2$^{4A}$ a reduction closer to 10-fold. So, despite the fact that the effects we observed in pulldown between recombinant Mre11 and Mer2$^{3A}$ and Mer2$^{4A}$ were not as striking the ones we detected in our Y2H analysis, we were able to confirm that Mer2 directly associates with Mre11, and that our mutant Mer2 proteins exhibit reduced Mre11 association.

DSB factors are recruited to chromosomes during meiotic prophase (*Panizza et al., 2011*), and we were interested in addressing whether the introduction of Mer2 mutants affected the chromosomal association of Mre11 and Rec114. We carried out immunofluorescence on meiotic chromosome spreads and analysed the signal corresponding to Rec114 or Mre11 (both carrying a COOH-terminal *13XMYC* tag) in cells expressing the different *MER2* alleles (*Figure 6—figure supplement 3A* and B). We did not detect any obvious differences in the levels of chromosomally associated Rec114 or Mre11 in cells expressing wildtype or mutant versions of Mer2. As such, while the interaction between Mer2 and Mre11 might be disrupted in the mutants, the ability of Mre11 (and Rec114) to localise to chromosomes during meiosis appeared unaffected.

We next were interested to shed more light on the interaction between Mer2 and Mre11, and to this end, we carried out XL-MS analysis on a complex of Mer2 and Mre11. These data revealed extensive cross-links across the length of Mre11 and Mer2, with the exception of the C-terminal region of Mer2 (*Figure 6D*). We paid particular attention to those cross-links that emanate from the residues that are in proximity of the Mer2$^{3A/4A}$ mutant region (amino acids 52–72) (*Figure 6D*, pale orange) and noted that these cross-links are predominantly formed with a region at the very C-terminus of Mre11. This region is predicted to be unstructured, but we noted that this region harbors a high-confidence SUMO interaction motif (SIM; residues 633–637) based on the SUMO-GPS prediction algorithm (*Zhao et al., 2014*). Intriguingly, a recent study on the role SUMOylation in meiosis (*Bhagwat et al., 2021*) reported Mer2 as a SUMO-target, and found that Mer2$^{K61}$ (which is one of the residues mutated in Mer2$^{3A}$) is directly SUMOylated. By comparing the location of the SIM (*Figure 6D*, green bar) with the Mer2-Mre11 cross-links, we noted that the SIM sits immediately adjacent to the most cross-linked region of Mre11. Taken together, we consider it a distinct possibility that the interaction between Mer2 and Mre11 might be influenced (and if so, likely enhanced) by SUMOylation of Mer2 during meiotic prophase.

Our earlier analysis of the phenotypes exhibited by cells expressing our Mer2 mutants, together with the fact that the MRX complex (including Mre11 specifically; *Johzuka and Ogawa, 1995*) is generally required for meiotic DSB formation (reviewed in *Borde, 2007*), prompted us to investigate meiotic DSB formation in *mer2-3a* and *mer2-4a* mutants. We used Southern blotting to track meiotic DSB formation at *YCR047C*, a confirmed DSB hotspot. (Note that we utilised the *sae2Δ* resection and repair-deficient background in order to enable DSB detection.) We detected a strong impairment of meiotic DSB activity specifically in *mer2-3a* or *mer2-4a*, to an extent that was comparable to what was

observed in *mer2Δ* strains (*Figure 6E*). Thus, Mer2[3A] and Mer2[4A] disrupt a key functionality of Mer2 that is required for meiotic DSB formation. Based on this and our earlier data, we propose that, in addition to its centrally positioned role in mediating chromosome axis and chromatin loop-tethering of the DSB machinery, a third key contribution that Mer2 (and its conserved N-terminal region) makes to enable DSB activity is establishing an interaction Mre11. While we cannot comprehensively rule out an alternative mechanism, our data suggest that not only the presence of Mre11 is required, but its specific interaction with Mer2 is essential for the initiation of DSB formation during meiotic prophase.

## Discussion

We have biochemically dissected the function of Mer2 in vitro to reveal novel features that have the potential to explain several in vivo characteristics of Spo11-dependent DSB formation. Firstly, we shed light on the interaction between Spp1 and Mer2. The interaction between Spp1 and Mer2 is tight (~25 nM) and not dependent on any post-translational modification or additional cofactors. As such, the interaction between Spp1 and Mer2 is likely constitutive in vivo. Importantly, we also find that Mer2 serves as a dimerisation platform for Spp1, effectively increasing its affinity for H3K4[me3] nucleosomes. Presumably this is essential given that the interaction between COMPASS bound Spp1 and H3K4[me3] is transient, whereas the association of the DSB forming machinery is an apparently more stable event (*Karányi et al., 2018*). Given that the Spp1 interaction domain of Mer2 is the same as the tetramerisation domain, we speculate that the antiparallel arrangement of coiled-coils of Mer2[core] (*Claeys Bouuaert et al., 2021*) form two oppositely oriented binding sites for Spp1, with V195 positioned at the centre (*Adam et al., 2018*; *Claeys Bouuaert et al., 2021*).

We find that Mer2 itself is a bona-fide nucleosome binder. This interaction occurs at high affinity, though we assume that the true affinity is somewhat less than what we determine from EMSA titrations. Given that Mer2 has been previously shown to bind DNA (*Claeys Bouuaert et al., 2021*), yet still binds to an NCP (with 147 bp DNA and no DNA overhangs), and that neither the loss of histone tails nor the acidic patch mutant disrupted the interaction, we suggest that Mer2 binds to nucleosomal DNA. This idea is supported by the XL-MS data on the Mer2-Spp1 complex bound to nucleosomes, which places Mer2 proximal to the DNA entry/exit site (*Figure 2D and E*). The tight and specific nucleosome binding ability of Mer2 provides a molecular basis for the observation that neither Spp1- nor Set1-mediated H3K4me3 are absolutely required to make meiotic DSBs (*Acquaviva et al., 2013*; *Sommermeyer et al., 2013*), unlike Mer2 itself (*Rockmill et al., 1995*). As such we speculate that in the absence of H3K4[me3] marks, or its reader Spp1, Mer2 binds stochastically to nucleosomes that are positioned in chromatin loops. Such a model speculates that some of these binding events present a nucleosome-depleted loop region to Spo11, but many do not, also explaining why meiotic DSBs are severely reduced in number in an *spp1Δ* or *set1Δ* background (*Acquaviva et al., 2013*; *Sommermeyer et al., 2013*). On the other hand, if Mer2 preferentially bound free DNA, as opposed to nucleosomal DNA, one might reasonably expect a less severe DSB phenotype in *spp1Δ* or *set1Δ* cells. The association between nucleosomes, Spp1 and Mer2, is important to establish the connection between chromosome axis-associated DSB factors and DSB sites that are localised in the loop (*Acquaviva et al., 2013*; *Sommermeyer et al., 2013*). We speculate that the combination of 'generic' nucleosome binding (via Mer2-nucleosomal DNA) and specific histone tail recognition (Spp1-PHD domain-H3 tail) endows the DSB machinery with the required binding strength and specificity.

Our observation and characterisation of the direct interaction between Hop1 and Mer2 offers a tantalising glimpse into the mechanism by which axial proteins may be recruiting DSB factors – through Mer2 – in order to regulate Spo11 activity. A key observation is that we only observed binding between Mer2 and Hop1[WT] in the presence of Red1 (*Figure 4A and B*). We could exclude any significant direct binding to Red1 (*Figure 4B*), which is consistent with the observations in mammals and fission yeast, where Hop1 orthologs have been implicated as the direct interactor of Mer2 orthologs (*Kariyazono et al., 2019*; *Stanzione et al., 2016*) – though we here, for the first time, demonstrate a direct interaction in vitro between purified components. We found that the use of a constitutively 'unlocked' Hop1 (Hop1[K593A]) leads to an increase in Mer2 binding, whereas the isolated C-terminus (Hop1[ΔHORMA]) does not show Mer2 interaction. These data clearly indicate a role for the HORMA domain in Mer2 association, but by which mechanism? Given that Hop1[WT] is in equilibrium between 'locked' and 'unlocked' as it binds and unbinds its own CM, one explanation could simply be that if Mer2 is incubated with Hop1[WT] for longer time periods in vitro then eventually a complex will form,

with presumably unlocked Hop1. In vivo unlocking is likely promoted by the interaction of the Red1 CM with the HORMA of Hop1. However, our in vitro pulldown experiments point towards an additional effect of Red1 on the Hop1-Mer2 binding. How might Red1 influence this binding? We did not detect direct binding to Red1 in a pulldown experiment. We therefore hypothesise several possibilities by which Red1 contributes to Mer2 binding to Hop1. The most simple explanation is that the CM of Red1 both unlocks Hop1, but also catalyses the closed conformation of the HORMA domain (*West et al., 2018*). Additionally, the oligomerisation effect of Red1 on Hop1 (Red1 is a tetramer; *West et al., 2019*) could promote Mer2 interaction by avidity. Alternatively, Red1 could (allosterically) influence the position of the HORMA domain. Our observation that the binding of Mer2 only apparently occurs with one 'face' of the HORMA domain suggests that the other 'face' might be directly oriented towards Red1. As such, Red1 might act as a conformational and stoichiometric enhancer of Mer2 binding, recruiting multiple Hop1 HORMAs and orienting them in three-dimensional constellation so that they are positioned favourably for interaction with Mer2.

Intriguingly, in fission yeast, the zinc finger of Hop1 (located in region corresponding with residues 348–364 within the C-terminal region of *Saccharomyces cerevisiae* Hop1) has been suggested to be required for Mer2 binding (*Kariyazono et al., 2019*). While we show that this region alone (which is now thought to be a PHD domain; *Ur and Corbett, 2021*) is not sufficient for Mer2 binding, it could nonetheless be involved in enhancing binding to Mer2. Regardless, our observations provide mechanistic insights into the specific recruitment of Mer2 to meiotic chromosomes. For example, our model might explain how Mer2 can specifically be recruited to meiotic chromosomes, and not form spurious interactions with non-chromosomal Hop1: non-chromosomal, monomeric Hop1 is thought to be largely present in the intramolecular 'closed' form (unless it is converted into the 'open/unbuckled' state by Pch2/TRIP13, which is thought to promote rapid chromosomal incorporation of Hop1) (*Cardoso da Silva and Vader, 2021*; *Raina and Vader, 2020*). Conversely, it might explain how DSB activity is negatively regulated by chromosome synapsis. Synapsis leads to recruitment of Pch2/TRIP13 and removal of Hop1 from the chromosomal axis (presumably by opening up Hop1 bound to Red1). This would be associated with immediate co-removal of Mer2, and once released Hop1 would transition into intramolecular Hop1, leading to disruption of the Mer2-Hop1 complex (*Chen et al., 2014*; *Vader, 2015*; *West et al., 2018*; *Raina and Vader, 2020*; *Subramanian et al., 2016*; *Yang et al., 2020*).

Finally, we attempted to create separation of function mutants for Mer2 by mutating a previously undescribed conserved region in the N-terminus of the protein. Both mutants in this region (Mer2[3A] and Mer2[4A]) were penetrant: they exhibited a loss of spore viability similar to cells lacking the Mer2 protein entirely (*Figure 5C*). Nonetheless, these mutants maintained an interaction with Hop1 in vitro and in vivo, and – in agreement with the proposed role of the Mer2-Hop1 interaction – exhibited normal localisation to the meiotic axis in vivo. In addition, although quantitative EMSAs showed that there was a reduction in affinity of Mer2[3A] for nucleosomes (*Figure 3—figure supplement 2C*), under physiological salt conditions (i.e. in the pulldown in *Figure 3F*) both Mer2 mutants could still form a complex with nucleosomes.

We discovered that our mutants interfere with an interaction between Mer2 and Mre11. We note that the observed severity of the effect of the Mer2[3A] and Mer2[4A] on the Mer2-Mre11 interaction was somewhat variable: Y2H analysis indicated a strong disruption of the interaction when using Mer2[3A] or Mer2[4A] compared to Mer2, and while we also observed interaction between Mer2 and Mre11 using in vitro purified proteins, and that the mutants reduced the interaction between Mer2 and Mre11, the difference between the mutants and wildtype was less pronounced. We infer that a factor/condition that strengthens the interaction between Mer2 and Mre11 is present in vegetatively growing yeast cells (i.e. the condition of Y2H analysis), but is lacking in our in vitro pulldown. This could be an additional protein factor, perhaps one of the other components of the Mre11-Rad50-Xrs2 complex, or a post-translational modification. For example, Mer2 has been previously shown to be phosphorylated by DDK and S-Cdk (*Henderson et al., 2006*; *Murakami and Keeney, 2014*), and Mer2[3A] or Mer2[4A] could potentially affect these phosphorylation events. Moreover, Mre11 has been previously shown to contain two SIMs. Since many meiotic proteins have been recently shown to be SUMOylated in yeast, among them Mer2, SUMOylation might serve as an important regulator (*Bhagwat et al., 2021*). Importantly, the residues that are mutated in Mer2[3A] are essentially universally conserved throughout the eukaryotic kingdom (*Figure 5—figure supplement 1*). As such, we surmise that the

direct interaction between Mer2 and Mre11, via these residues, is a universal feature of the meiotic program. Intriguingly, the Mer2[3A] and Mer2[4A] mutants do not disrupt the chromosomal localisation of Mre11 during meiosis (*Figure 6—figure supplement 3B*). As such this suggests that rather than a simple disruption of localisation, the Mer2 mutants might be disrupting an allosteric effect of Mer2 on Mre11. Alternatively, the efficient co-localisation of Mre11 and Mer2 into DSB foci might be disrupted (as might the co-localisation of additional cofactors). We note that several other DSB factors have been shown via Y2H to interact with Mer2 (*Arora et al., 2004*). An important future research goal should be to comprehensively biochemically anaylse the interactions between Mer2 and the DSB machinery, also in light of our findings here.

Taken together our data show that Mer2 forms the keystone of meiotic recombination, binding directly to the axis – via Hop1-, and to the loop – via nucleosomes. Presumably, once assembled on the loop axis, Mer2 establishes interactions with Rec114 and Mei4 in phospho-dependent manner (likely via the PH domains of Rec114; *Boekhout et al., 2019*; *Li et al., 2006*; *Panizza et al., 2011*), an association that may also be somehow further controlled by liquid-liquid phase separation (*Claeys Bouuaert et al., 2021*). Simultaneously, our data indicate that the N-terminus of Mer2 recruits the MRX complex via Mre11, a step that we suggest is critical for the formation of meiotic DSBs and the successful completion of meiosis. In organisms with a synaptonemal complex (SC), downregulation of meiotic DSB formation is associated with chromosome synapsis. It has been shown that synapsis results in the Pch2[TRIP13]-mediated displacement of Hop1 from the axis (*Börner et al., 2008*; *Chen et al., 2014*; *Joshi et al., 2009*; *Lambing et al., 2015*; *San-Segundo and Roeder, 1999*; *Subramanian et al., 2016*; *Wojtasz et al., 2009*). Based on our work, we posit that this action would also result in the displacement of the crucial DSB factor Mer2 from the axis (as removed Hop1 would no longer be bound to Red1, thus weakening its affinity for Mer2). Additionally, once released, Hop1 would be expected to re-bind its own CM, thus 'snapping shut' and preventing re-binding to Red1, and also presumably preventing re-recruitment of Mer2. Together, these effects provide a potential molecular rationale for the functional connection between chromosome synapsis and reduced DSB activity.

Our findings highlight the power of biochemical reconstitution in dissecting the function of complex biological systems. Mer2 emerges as the keystone of meiotic recombination, establishing interactions with the chromosome axis, nucleosomes, and the DSB machinery (*Figure 6—figure supplement 4*). Larger and more ambitious reconstitutions will enable us to probe the role of additional protein cofactors and post-translational modifications in meiotic regulation.

## Materials and methods
### Protein expression and purification

Sequences of *S. cerevisiae SPP1*, *HOP1*, *RED1*, and *MRE11* were derived from SK1 strain genomic DNA. Due to the presence of an intron in *MER2,* this was amplified as two separate fragments and Gibson assembled together.

All Mer2 constructs were expressed as an 3C HRV cleavable N-terminal MBP fusion in chemically competent C41 *E. coli* cells. Protein expression was induced by addition of 250 µM IPTG and the expression continued at 18°C overnight. Cells were washed with 1× PBS and resuspended in lysis buffer (50 mM HEPES pH 7.5, 300 mM NaCl, 5% glycerol, 0.1% Triton-X 100, 1 mM MgCl$_2$, 5 mM β-mercaptoethanol). Resuspended cells were lysed using an EmulsiFlex C3 (Avestin) in the presence of DNAse (10 µg/mL) and AEBSF (25 µg/mL) before clearance at 20,000 *g* at 4°C for 30 min. Cleared lysate was applied on a 5 mL MBP-trap column (GE Healthcare) and extensively washed with lysis buffer. Mer2 constructs were eluted with a lysis buffer containing 1 mM maltose and passed through a 6 mL ResourceQ column (GE Healthcare) equilibrated in 50 mM HEPES pH 7.5, 100 mM NaCl, 5% glycerol, 5 mM β-mercaptoethanol. The proteins were eluted by increasing salt gradient to 1 M NaCl. Protein containing elution fractions were concentrated on Amicon concentrator (100 kDa MWCO) and loaded a Superose 6 16/600 (GE Healthcare) pre-equilibrated in SEC buffer (50 mM HEPES pH 7.5, 300 mM NaCl, 10% glycerol, 1 mM TCEP). Untagged Mer2[FL] was prepared likewise until concentration of protein eluted from ResourceQ. The concentrated eluent was mixed with 3C HRV protease in a molar ratio of 50:1 and incubated at 4°C for 6 hr. Afterwards, the cleaved protein was loaded on a Superose 6 16/600 pre-equilibrated in SEC buffer for cleaved Mer2 (20 mM HEPES pH 7.5, 500 mM NaCl, 10% glycerol, 1 mM TCEP, 1 mM EDTA, AEBSF).

Spp1 constructs were produced as an 3C HRV cleavable N-terminal MBP or GST fusion in a similar manner as MBP-Mer2. To purify GST-Spp1, cleared lysate was applied on GST-Trap (GE Healthcare) before extensive washing with lysis buffer. The protein was eluted with a lysis buffer with 40 mM reduced glutathione and passed through ResourceQ. Both GST and MBP could be cleaved by adding 3C HRV protease to concentrated protein (using an Amicon concentrator with 30 kDa cutoff) in 1:50 molar ratio. After an ~6 hr incubation at 4°C, the cleaved protein was loaded on Superdex 200 16/600 pre-equilibrated in SEC buffer (50 mM HEPES pH 7.5, 300 mM NaCl, 10% glycerol, 1 mM TCEP).

Hop1 constructs were produced as 3C HRV cleavable N-terminal Twin-StrepII tag in BL21 STAR *E. coli* cells. The expression was induced by addition of 250 µM IPTG and the expression continued at 18°C for 16 hr. Cleared lysate was applied on Strep-Tactin Superflow Cartridge (Qiagen) before extensive washing in lysis buffer. The bound protein was eluted with a lysis buffer containing 2.5 mM desthiobiotin and loaded on HiTrap Heparin HP column (GE Healthcare) and subsequently eluted with increasing salt gradient to 1 M NaCl. Eluted Strep-Hop1 constructs were concentrated on a 30 kDa MWCO Amicon concentrator and loaded on Superdex 200 16/600 pre-equilibrated in SEC buffer.

Red1 was produced in insect cells as a C-terminal MBP-fusion either alone or co-expressed with Strep-Hop1. Bacmids were in both cases produced in EmBacY cells and subsequently used to trans-fect Sf9 cells to produce baculovirus. Amplified baculovirus was used to infect Sf9 cells in 1:100 dilu-tion prior to 72 hr cultivation and harvest. Cells were extensively washed and resuspended in Red1 lysis buffer (50 mM HEPES pH 7.5, 300 mM NaCl, 10% glycerol, 1 mM MgCl$_2$, 5 mM β-mercaptoeth-anol, 0.1% Triton-100). Resuspended cells were lysed by sonication in the presence of Benzonase and a protease inhibitor cocktail (Serva) before clearance at 40,000 *g* at 4°C for 1 hr. Cleared lysate was loaded on Strep-Tactin Superflow Cartridge (in case of Red1-Hop1 complex) or MBP-trap column (in case of Red1 alone). Proteins were eluted using a lysis buffer containing 2.5 mM desthiobiotin and 1 mM maltose, respectively. Partially purified proteins were further passed through HiTrap Heparin HP column and eluted with increasing salt gradient to 1 M NaCl. Purified proteins were subsequently concentrated using Pierce concentrator with 30 kDa cutoff in 50 mM HEPES pH 7.5, 300 mM NaCl, 10% glycerol, 1 mM TCEP. Because of the small yield of the proteins, the SEC purification step was neglected and the purity of the proteins was checked using the Refeyn One mass photometer.

Mre11 was produced as a C-terminal Twin-StrepII tag in insect cells using the same expression conditions as for Red1 protein. The cell pellet was resuspended in Mre11 lysis buffer (50 mM HEPES pH 7.5, 300 mM NaCl, 5% glycerol, 0.01% NP40, 5 mM β-mercaptoethanol, AEBSF, Serva inhibitors). Resuspended cells were lysed by sonication before clearance at 40,000 *g* at 4°C for 1 hr. Cleared lysate was loaded on a 5 mL Strep-Tactin XT Superflow Cartridge (IBA) followed by first wash using 25 mL of high-salt wash buffer (20 mM HEPES pH 7.5, 500 mM NaCl, 5% glycerol, 0.01% NP40, 1 mM β-mercaptoethanol) and second wash step using 25 mL of low-salt wash buffer (20 mM HEPES pH 7.5, 150 mM NaCl, 5% glycerol, 0.01% NP40, 1 mM β-mercaptoethanol). The Mre11 protein was eluted with 50 mL of low-salt wash buffer containing 50 mM biotin. Partially purified protein was further loaded onto a 5 mL Heparin column (GE Healthcare) pre-equilibrated in a low-salt wash buffer and eluted with increasing salt gradient to 1 M NaCl. The fractions containing Mre11 protein were concen-trated on a 50 kDa MWCO Amicon concentrator and applied onto a Superdex 200 10/300 column (GE Healthcare) pre-equilibrated in Mre11 SEC buffer (20 mM HEPES pH 7.5, 300 mM NaCl, 5% glycerol, 1 mM β-mercaptoethanol, 1 mM TCEP).

## SEC-MALS analysis

Fifty µL samples at 5–10 µM concentration were loaded onto a Superose 6 5/150 analytical size exclu-sion column (GE Healthcare) equilibrated in buffer containing 50 mM HEPES pH 7.5, 1 mM TCEP, 300 mM NaCl (for samples without nucleosomes) or 150 mM NaCl (for samples with nucleosomes) attached to an 1260 Infinity II LC System (Agilent). MALS was carried out using a Wyatt DAWN detector attached in line with the size exclusion column.

## Microscale thermophoresis

Triplicates of MST analysis were performed in 50 mM HEPES pH 7.5, 300 mM NaCl, 5% glycerol, 1 mM TCEP, 0.005% Tween-20 at 20°C. The final reaction included 20 µM RED-NHS labelled untagged Spp1 (labelling was performed as in manufacturer's protocol – Nanotemper) and titration series of

MBP-Mer2 constructs (concentrations calculated based on oligomerisation stage of Mer2). The final curves were automatically fitted in Nanotemper analysis software.

## Recombinant nucleosome production

Recombinant *Xenopus laevis* histones were purchased from 'The Histone Source' (Colorado State) with the exception of H3-C110A_K4C cloned into pET3, which was kindly gifted by Francesca Matirolli. The trimethylated H3 in C110A background was prepared as previously described (*Simon et al., 2007*). *X. laevis* histone expression, purification, octamer refolding, and mononucleosome reconstitution were performed as described (*Luger et al., 1999*). Plasmids for the production of 601–147 (pUC19) and 601–167 (pUC18) DNA were kindly gifted by Francesca Matirolli (Hubrecht Institute, Utrecht) and Andrea Musacchio (MPI Dortmund), respectively. DNA production was performed as previously described (*Luger et al., 1999*). Reconstituted mononucleosomes were shifted to 20 mM Tris pH 7.5, 150 mM NaCl, 1 mM EDTA, 1 mM TCEP with addition of 20% glycerol prior to freezing in –80°C.

## Electrophoretic mobility shift assays

Quadruplicate EMSAs were carried out as previously described (*Weir et al., 2016*), at a constant nucleosome/NCP/DNA concentration of 10 nM with the DNA being post-stained with SYBRGold (Invitrogen). Gels were imaged using a ChemiDocMP (Bio-Rad Inc). Nucleosome depletion in each lane was quantitated by ImageJ, using measurements of triplicate of the nucleosome alone for each individual gel as a baseline. Binding curves were fitted using Prism software and the following algorithm ($Y = Bmax*X^h/(K_D^h + X^h)$). It was necessary in each Mer2 case to add a Hill coefficient to obtain the best fit.

## Streptactin pulldown

Streptactin pulldowns were performed using pre-blocked streptavidin magnetic beads (Pierce) in a pulldown buffer (20 mM HEPES pH 7.5, 300 mM NaCl, 5% glycerol, 1 mM TCEP). One µM Strep-Hop1 or Mre11-Strep as a bait was incubated with 3 µM Mer2 as a prey in 40 µL reaction for 2 hr on ice without beads and another 30 min after addition of 10 µL of beads pre-blocked with 1 mg/mL BSA for 2 hr. After incubation, the beads were washed twice with 200 µL of buffer before elution of the proteins with a 1× Laemmli buffer. Samples were loaded on 10% SDS-PAGE gel and afterwards stained with InstantBlue.

## Amylose pulldown

Amylose pulldowns were performed using pre-blocked Amylose beads (New England BioLabs) in a pulldown buffer (20 mM HEPES pH 7.5, 300 mM NaCl, 5% glycerol, 1 mM TCEP). One µM Red$^{I743R}$-MBP or Red$^{I743R}$-MBP/Strep-Hop1 as a bait was incubated with 3 µM Mer2 as a prey in 40 µL reaction for 2 hr on ice without beads and another 1 hr after addition of 10 µL of beads pre-blocked with 1 mg/mL BSA for 2 hr. After incubation, the beads were washed twice with 200 µL of buffer before elution of the proteins with a buffer containing 1 mM maltose. Samples were loaded on 10% SDS-PAGE gel and afterwards stained with InstantBlue.

## Biotinylated nucleosome pulldown

Biotinylated nucleosomes (0.5 µM) or NCP (0.4 µM) were incubated with prey proteins (1.5 µM) for 30 min on ice in buffer containing 20 mM HEPES pH 7.5, 150 mM NaCl, 5% glycerol, 1 mM EDTA, 0.05% Triton-X100, 1 mM TCEP in a reaction volume of 40 µL. Ten µL of protein mix were taken as an input before adding 10 µL of pre-equilibrated magnetic Dynabeads M 270 streptavidin beads (Thermo Fisher Scientific) to the reaction. The samples with beads were incubated on ice for 2 min before applying magnet and removing the supernatant. The beads were washed twice with 200 µL of buffer. To release the streptavidin from the beads, Laemmli buffer (1×) was added to the beads and incubated for 10 min. Samples were analysed on 10–20% SDS-PAGE gel and stained by InstantBlue.

## Analytical SEC

Analytical SEC was performed using Superose 6 5/150 GL column (GE Healthcare) in a buffer containing 20 mM HEPES pH 7.5, 150 mM NaCl, 5% glycerol, 1 mM TCEP, 1 mM EDTA. All samples

were eluted under isocratic elution at a flow rate of 0.15 mL/min. Protein elution was monitored at 280 nm. Fractions were subsequently analysed by SDS-PAGE and InstantBlue staining. To detect complex formation, proteins were mixed at 5 µM concentration in 50 µL and incubated on ice for 1 hr prior to SEC analysis.

## Mass photometry

MP was performed in 20 mM HEPES pH 7.5, 150 mM NaCl, 5% glycerol, 1 mM TCEP, 1 mM EDTA. Mer2 and mononucleosomes (600 nM) were mixed and incubated for 1 hr on ice prior to analysis using the Refeyn One mass photometer. Immediately before analysis, the sample was diluted 1:10 with the aforementioned buffer. Molecular mass was determined in Analysis software provided by the manufacturer using a NativeMark- (Invitrogen) based standard curve created under the identical buffer composition.

## Cross-linking mass spectrometry

For XL-MS analysis proteins were dissolved in 200 µL of 30 mM HEPES pH 7.5, 1 mM TCEP, 300 mM NaCl (for samples without nucleosomes) or 150 mM NaCl (for samples with nucleosomes) to final concentration of 3 µM, mixed with 3 µL of DSBU (200 mM) and incubated for 1 hr in 25°C. The reaction was quenched by addition of 20 µL of Tris pH 8.0 (1 M) and incubated for another 30 min in 25°C. The crosslinked sample was precipitated by addition of 4× volumes of 100% cold acetone ON in –20°C and subsequently analysed as previously described (*Pan et al., 2018*).

## Yeast strains

All strains, except those used for Y2H analysis, are of the SK1 background. See Supplementary material for a description of genotypes of strains used per experiment. For *MER2* alleles, constructs containing *pMER2*(−1000–1), the coding sequence of *MER2* lacking its intron (i.e. wildtype, or 3a/4a-containing sequences), C-terminal 3HA tag, and 500 base pairs of downstream sequence flanked by HindIII and NarI restriction enzymes were custom-synthesised by Genewiz Inc These constructs were recloned in a YIPlac128 plasmid carrying *LEU2* using restriction cloning. These plasmids were integrated at *pMER2* in front of *mer2::HISMX6* following EcoRI linearisation. Correct single copy integration was confirmed by PCR.

## Growth conditions for synchronous meiosis

Yeast strains were patched onto YP-Glycerol plates and transferred to YP-4%Dextrose plates. After this, cells were grown overnight in liquid YPD culture (room temperature) followed by inoculation in pre-sporulation media (BYTA; 50 mM Sodium Phthalate-buffered, 1% yeast extract, 2% tryptone, and 1% acetate) at $OD_{600}$ = 0.3. Cells were grown for 18 hr in BYTA at 30°C, washed twice in water and resuspended in sporulation media (0.3% potassium acetate) at $OD_{600}$ = 1.9 to induce meiosis at 30°C.

## Spore viability and efficiency

Cells were synchronously induced into meiosis and incubated for 24 hr. Of each strain, 200 cells were counted using a standard bright-field microscope, and monads, dyads, and tetrads were scored. For viability, the indicated number of tetrads was dissected using standard manipulation methods and grown on YPD plates. Spore viability was calculated as a percentage of the total number of viable spores.

## Surface spreading of chromosomes and immunofluorescence

Two mL of meiotic cells (t = 3 hr induction) were collected at indicated time points, killed by addition of 1% sodium azide and processed. Cells were treated with 200 mM Tris pH 7.5, 20 mM dithiothreitol for 2 min at room temperature followed by spheroplasting at 30°C in 1 M sorbitol, 2% potassium acetate, 0.13 µg/µL zymolyase (20 min). Spheroplasts were washed two times in 1 mL ice-cold MES-Sorbitol solution (1 M sorbitol, 0.1 M MES pH 6.4, 1 mM EDTA, 0.5 mM MgCl₂) and resuspended in 55 µL of MES-Sorbitol. Twenty µL of spheroplasts were placed on clean glass slides (that were dipped in ethanol overnight and air-dried) and 2× volumes of fixing solution (3% paraformaldehyde, 3.4% sucrose) were added. This was followed by addition of four volumes of 1% Lipsol, and mixing through gentle rotation. After 1 min, 4× volumes of fixing solution were added. A glass rod was used to

mechanically spread chromosomes, after which samples were dried overnight at room temperature, and stored at −20°C. Slides were treated with 0.4% Photoflo (Kodak)/PBS for 3 min, after which slides were dipped in PBS with gentle shaking (5 min). Samples were blocked by incubation with 5% BSA in PBS for 15 min at room temperature. Overnight incubation with desired primary antibodies was performed in a humidified chamber at 4°C, after which slides were subjected to 2× washes of 10 min in PBS with gentle shaking followed by incubation with fluorescent-conjugated secondary antibody for 3 hr at room temperature. The slides were washed 2× and mounted using 20 µL of Vectashield mounting solution containing 4′,6-diamidine-2′-phenylindole dihydrochloride (DAPI) (Vector Laboratories). Chromosome surface spreads were immunostained with rat α-HA at 1:200 (Roche), mouse α-MYC (9E10) at 1:200, and rabbit α-Hop1 at 1:200 (home made). Hop1 antibody production was performed at the antibody facility of the Max-Planck-Institute of Molecular Cell Biology and Genetics (Dresden, Germany) using affinity purified full length 6xHis-tagged Hop1. Secondary antibodies were used at the following concentrations: Alexa 488-conjugated donkey α-rat at 1:500; Texas Red 594-conjugated donkey α-rabbit at 1:500, and Cy3-conjugated donkey α-mouse at 1:500. Image acquisition was done by obtaining serial z-stacks of 0.2 µm thickness at room temperature using 100 × 1.42 NA PlanApo-N objective (Olympus) on a DeltaVision imaging system (GE Healthcare) equipped with an sCMOS camera (PCO Edge 5.5). The z-stack images were deconvolved using SoftWoRx software. Quantifications for the number of foci of Mer2$^{WT}$-3HA, Mer2$^{3A}$-3HA Mer2$^{4A}$-HA were done using the 'Spots' function of the Imaris software (Bitplane). Prism 8 (GraphPad) was used to generate the Scatter plots. Statistical significance was assessed by performing Mann-Whitney U-test. For representative images, Fiji/ImageJ software was used to obtain maximum intensity projection images.

## Co-immunoprecipitations

One-hundred mL of meiotic cultures (at 4 hr into a meiotic time course) were harvested by spinning down at 3000 rpm for 3 min. Samples were washed with cold H$_2$O and snap frozen. To the pelleted cells, the following was added: 300 µL of ice-cold IP buffer (consisting of 50 mM Tris–HCl pH 7.5, 150 mM NaCl, 1% Triton X-100 and 1 mM EDTA pH 8.0, and a cocktail of protease inhibitors which was freshly added) and acid-washed glass beads. A FastPrep-24 disruptor (MP Biomedicals) was used to break the cells (MP Biomedicals) (setting: two 50 s cycles (speed 6)). Lysates were spun 30 s at 500 rpm, after which the supernatant was transferred to a 15 mL falcon tube. The resulting lysate was sonicated for 25 cycles (30 s on/30 s off, high power range) using a Bioruptor-Plus sonicator (Diagenode), and subsequently spun down 30 min at maximum speed at 4°C. The supernatant was transferred into new microcentrifuge tubes, and 10% of the supernatant (~50 µL) was collected as input. One µL of antibody (α-HA; BioLegend) was added, and the lysate was rotated for 3 hr at 4°C; 30 µL of buffer-washed Dynabeads protein G (Invitrogen, Thermo Fisher Scientific) was subsequently added, and the lysate was rotated overnight at 4°C. The next day, Dynabeads were washed four times with 500 µL of ice-cold IP buffer. For the final wash, beads were transferred to a new microcentrifuge tube. After removal of the supernatant, the beads were resuspended in 40 µL IP buffer, and 20 µL of 3XSDS-loading buffer was added. Samples were incubated for 5 min at 95°C. For the inputs, the following protocol was used: Inputs were precipitated by the addition of 5 µL 100% trichloroacetic acid (TCA) (10% final concentration), and incubated on ice for 30 min. Precipitates were collected by centrifugation, for 1 min at maximum speed, and washed with ice-cold acetone. Precipitations were dried and resuspended in resuspension buffer (40 µL, 50 mM Tris-HCl 7.5, and 6 M urea) for 30 min (on ice). Dissolution was encouraged by pipetting and vortexing. Ten µL of 5XSDS-loading buffer was added, and samples were incubated for 5 min at 95°C.

## Western blot analysis

For Western blot analysis, protein lysates from yeast meiotic cultures were prepared using TCA precipitation and run on 8% or 10% SDS gels, transferred for 90 min at 300 mA and blotted with the selected antibodies, as described (*Kuhl et al., 2020*). Primary antibodies with respective dilutions were used: rabbit α-Hop1 (made in-house; 1:10,000), rabbit α-Mer2$^{40-271}$ (made in-house; 1:10,000); mouse α-Pgk1 (Thermo Fisher, 1:5000); rabbit α-phospho-Histone-H3-Thr11 (Abcam, 1:1000), mouse α-HA (Biolegend, diluted 1:500), α-Myc-9E11 (Abcam, ab56, 1:1000).

## Southern blot analysis

For Southern blot assay, DNA from meiotic samples was prepared as described (*Vader et al., 2011*). DNA was digested with HindIII (to detect DSBs at the control *YCR047C* hotspot) followed by gel electrophoresis, blotting of the membranes and radioactive (32P) hybridisation using a probe for *YCR047C* (chromosome *III*; 209,361–201,030) (*Raina and Vader, 2020*). DSBs signals were monitored by exposure of an X-ray film which was analysed using a Typhoon Trio scanner (GE Healthcare) after 1 week.

## Yeast-2-ybrid

*MER2* variants and their potential interactors were cloned into pGAD-C1 or pGBDU-C1 vectors, respectively. The resulting plasmids were co-transformed into the *S. cerevisiae* reporter strain (yWL365) and plated onto the selective medium lacking leucine and uracil. For drop assay, 2.5 µL from 10-fold serial dilutions of cell cultures with the initial optical density (OD$_{600}$) of 0.5 were spotted onto -Leu/-Ura (control) and -Leu/-Ura/-His plates. Cells were grown at 30°C for up to 7 days and imaged at time points indicated in the figures.

## IP-mass Spectrometry

Cells (450 mL) were harvested after 3 hr of synchronised meiosis. Three independent cultures were prepared for each strain. Pellets were lysed in cryo-mill and the resulting powder was resuspended in 25 mL of lysis buffer (50 mM HEPES pH 7.5, 300 mM NaCl, 5% glycerol, 0.01% NP40, 5 mM β-mercaptoethanol, AEBSF, Serva protease inhibitors, 1 mM NEM, 1× Complete Mini EDTA-Free protease inhibitors; Roche). The lysate was cleared by centrifugation at 10,000 rpm for 1 hr. The supernatant was incubated with 5 µL of α-HA antibody (Sigma-Aldrich, H6908) for 3 hr at 4°C followed by the addition of 25 µL magnetic Dynabeads Protein G (Invitrogen, 10,004D) pre-equilibrated with wash buffer (20 mM HEPES pH 7.5, 100 mM NaCl, 5% glycerol, 0.01% NP40, 1 mM β-mercaptoethanol). The samples were incubated overnight at 4°C. Next day, the samples were centrifuged at 2500 rpm for 1 min at 4°C and the beads were washed six times with 150 µL of 20 mM ammonium bicarbonate. Proteins were eluted with 75 µL of 2× SDS Laemmli buffer and boiled for 5 min at 95°C. The samples were analysed by Proteomics Core Facility (EMBL, Heidelberg).

## Yeast strains used in this study

| Yeast strain | Genotype | Used in figure |
|---|---|---|
| yGV8 | MATa,ho::LYS2, lys2, ura3, leu2::hisG, his3::hisG, trp1::hisG<br>MATalpha, ho::LYS2, lys2, ura3, leu2::hisG, his3::hisG, trp1::hisG | 5B–D |
| yWL365 | MATa,ura3-52, leu2-3, his3, trp1, gal4Δ, gal80Δ, GAL2-ADE2, LYS2::GAL1-HIS3, met2::GAL7-lacZ, | 6A, S10 |
| yGV1129 | MATa, ho::LYS2, TRP1, his3::hisG, ura3, lys2, leu2::hisG,REC114-13MYC::HIS3<br>MATalpha, ho::LYS2, trp1::hisG, his3::hisG, URA3, lys2, LEU2,REC114-13MYC::HIS3 | S11B |
| yGV1375 | MATa, ho::LYS2, TRP, his3::hisG, ura3, LEU2, MRE11-13MYC::HIS3<br>MATalpha, ho::LYS2, trp1::hisG, his3::hisG, URA3, leu2::hisG,MRE11-13MYC::HIS3 | S11A |
| yGV4744 | MATa, ho::LYS2, lys2, ura3, leu2::hisG, his3::hisG, trp1::hisG, sae2Δ::LEU2<br>MATalpha, ho::LYS2, lys2, ura3, leu2::hisG, his3::hisG, trp1::hisG, sae2Δ::LEU2 | 6C |
| yGV4874 | MATa, ho::LYS2, lys2, ura3, leu2::hisG, his3::hisG, trp1::hisG, mer2Δ::HISMX6::pMER2-MER2-3HA::LEU2<br>MATalpha, ho::LYS2, lys2, ura3, leu2::hisG, his3::hisG, trp1::hisG, mer2Δ::HISMX6::pMER2-MER2-3HA::LEU2 | 5B–E, S12A,B |
| yGV4879 | MATa, ho::LYS2, lys2, ura3, leu2::hisG, his3::hisG, trp1::hisG, mer2Δ::HISMX6::pMER2-Mer2-4A-3HA::LEU2<br>MATalpha, ho::LYS2, lys2, ura3, leu2::hisG, his3::hisG, trp1::hisG, mer2Δ::HISMX6::pMER2-Mer2-4A-3HA::LEU2 | 5B–E |

*Continued on next page*

*Continued*

| Yeast strain | Genotype | Used in figure |
|---|---|---|
| yGV4889 | MATa, ho::LYS2, lys2, ura3, leu2::hisG, his3::hisG, trp1::hisG, mer2Δ::HISMX6<br>MATalpha, ho::LYS2, lys2, ura3, leu2::hisG, his3::hisG, trp1::hisG, mer2Δ::HISMX | 5B,C |
| yGV4913 | MATa, ho::LYS2, lys2, ura3, leu2::hisG, his3::hisG, trp1::hisG, mer2Δ::HISMX6, sae2Δ::LEU2<br>MATalpha, ho::LYS2, lys2, ura3, leu2::hisG, his3::hisG, trp1::hisG, mer2Δ::HISMX6, sae2Δ::LEU2 | 6C |
| yGV4931 | MATa, ho::LYS2, lys2, ura3, leu2::hisG, his3::hisG, trp1::hisG, mer2Δ::HISMX6::pMER2-mer2-4A-3HA::LEU2, sae2Δ::LEU2<br>MATalpha, ho::LYS2, lys2, ura3, leu2::hisG, his3::hisG, trp1::hisG,mer2Δ::HISMX6::pMER2-mer2-4A-3HA::LEU2, sae2Δ::LEU2 | 6C |
| yGV4933 | MATa, ho::LYS2, lys2, ura3, leu2::hisG, his3::hisG, trp1::hisG, mer2Δ::HISMX6::pMER2-mer2-3A-3HA::LEU2<br>MATalpha, ho::LYS2, lys2, ura3, leu2::hisG, his3::hisG, trp1::hisG, mer2Δ::HISMX6::pMER2-mer2-3A-3HA::LEU2 | 5B–E |
| yGV4934 | MATalpha, ho::LYS2, lys2, ura3, leu2::hisG, his3::hisG, trp1::hisG, mer2Δ::HISMX6::pMER2-mer2-3A-3HA::LEU2, sae2Δ::LEU2<br>MATa, ho::LYS2, lys2, ura3, leu2::hisG, his3::hisG, trp1::hisG, mer2Δ::HISMX6::pMER2-mer2-3A-3HA::LEU2, sae2Δ::LEU2 | 6C |
| yGV4957 | MATa, ho::LYS2, lys2, ura3, leu2::hisG, his3::hisG, trp1::hisG, mer2Δ::HISMX6::pMER2-mer2-3HA::LEU2, sae2Δ::LEU2<br>MATalpha, ho::LYS2, lys2, ura3, leu2::hisG, his3::hisG, trp1::hisG,mer2Δ::HISMX6::pMER2-mer2-3HA::LEU2, sae2Δ::LEU2 | 6C |
| yGV5052 | MATa, ho::LYS2, lys2, URA3, leu2::hisG, his3::hisG, trp1::hisG, mer2Δ::HISMX6::pMER2-Mer2-3HA::LEU2, MRE11-13MYC::HIS3<br>MATalpha, ho::LYS2, lys2, ura3, leu2::hisG, his3::hisG, trp1::hisG, mer2Δ::HISMX6::pMER2-Mer2-3HA::LEU2, MRE11-13MYC::HIS3 | S11A, S12B |
| yGV5053 | MATa, ho::LYS2, trp1::hisG, his3::hisG, URA3, leu2::hisG MRE11-13MYC::HIS3, rmer2Δ::HISMX6::pMER2-Mer2-3A-3HA::LEU2<br>MATalpha, ho::LYS2, trp1::hisG, his3::hisG, URA3, leu2::hisG MRE11-13MYC::HIS3, mer2Δ::HISMX6::pMER2-Mer2-3A-3HA::LEU2 | S11A, S12B |
| yGV5054 | MATa, ho::LYS2, lys2, URA3, leu2::hisG, his3::hisG, trp1::hisG, mer2Δ::HISMX6::pMER2-Mer2-3HA::LEU2, REC114-13MYC::HIS3<br>MATalpha, ho::LYS2, lys2, URA3, leu2::hisG, his3::hisG, trp1::hisG,mer2Δ::HISMX6::pMER2-Mer2-3HA::LEU2, REC114-13MYC::HIS3 | S11B, S12A |
| yGV5055 | MATalpha, ho::LYS2, trp1::hisG, his3::hisG, URA3, lys2, LEU2, REC114-13MYC::HIS3, mer2Δ::HISMX6::pMER2-Mer2-4A-3HA::LEU2<br>MATa, ho::LYS2, trp1, his3::hisG, ura3, lys2, leu2::hisG, REC114-13MYC::HIS3, mer2Δ::HISMX6::pMER2-Mer2-4A-3HA::LEU2 | S11B, S12A |
| yGV5056 | MATalpha, ho::LYS2, lys2, ura3, leu2::hisG, his3::hisG, trp1::hisG,mer2Δ::HISMX6::pMER2-Mer2-3A-3HA::LEU2, REC114-13MYC::HIS3<br>MATa, ho::LYS2, lys2, ura3, leu2::hisG, his3::hisG, trp1::hisG, mer2Δ::HISMX6::pMER2-Mer2-3A-3HA::LEU2, REC114-13MYC::HIS3 | S11B, S12A |
| yGV5057 | MATa, ho::LYS2, lys2, ura3, leu2::hisG, his3::hisG, trp1::hisG, mer2Δ::HISMX6::pMER2-Mer2-4A-3HA::LEU2, MRE11-13MYC::HIS3<br>MATalpha, ho::LYS2, trp1::hisG, his3::hisG, URA3, leu2::hisG MRE11-13MYC::HIS3, mer2Δ::HISMX6::pMER2-Mer2-4A-3HA::LEU2 | S11A, S12B |

*Continued on next page*

*Continued*

| Yeast strain | Genotype | Used in figure |
|---|---|---|
| yWL419 | *MATa, ho::LYS2, lys2, ura3, leu2::hisG, TRP, his4B::LEU2 spo11-Y135F::KanMX4 rec107(mer2)delete::HISMX6::pMer2-Mer2-3HA::LEU2*<br>*MATalpha, ho::LYS2, lys2, ura3, leu2::hisG, TRP, his4B::LEU2 spo11-Y135F::KanMX4 rec107(mer2)delete::HISMX6::pMer2-Mer2-3HA::LEU2* | S9, 5F, 5G |
| yWL420 | *MATa, ho::LYS2, lys2, ura3, leu2::hisG, TRP, his4B::LEU2 spo11-Y135F::KanMX4 rec107(mer2)delete::HISMX6::pMer2-Mer2-4A-3HA::LEU2*<br>*MATalpha, ho::LYS2, lys2, ura3, leu2::hisG, TRP, his4B::LEU2 spo11-Y135F::KanMX4 rec107(mer2)delete::HISMX6::pMer2-Mer2-4A-3HA::LEU2* | S9, 5F, 5G |
| yWL411 | *MATa, ho::LYS2, lys2, ura3, leu2::hisG, TRP, his4B::LEU2 spo11-Y135F::KanMX4*<br>*MATalpha, ho::LYS2, lys2, ura3, leu2::hisG, TRP, his4X::LEU2-(Bam)-URA3 spo11-Y135F::KanMX4* | S9, 5F, 5G |

## Expression constructs used in this study

| Plasmid ID | Description | Reference |
|---|---|---|
| pWL284 | MBP-Spp1 FL | This study |
| pWL 341 | GST-Spp1 FL | This study |
| pWL146 | MBP-Mer2 FL | This study |
| pWL186 | MBP-Mer2 140–256 (Core) | This study |
| pWL868 | MBP-Mer2 1–256 (ΔC) | This study |
| pWL1330 | MBP-Mer2 140C (ΔN) | This study |
| pWL939 | MBP-Mer2 1–139 | This study |
| pWL1077 | MBP-Mer2 255C | This study |
| pWL654 | *X. laevis* H3 C110A, K4C | This study |
| pWL658 (for *E. coli*) pWL661 (for insect cells) | Strep-Hop1 FL | This study |
| pWL1104 | Strep-Hop1 256C (ΔHORMA) | This study |
| pWL1139 | Strep-Hop1 K593A (CM mutant) | This study |
| pWL1157 | Red1-MBP_I743R | This study |
| pWL275 | Mre11-Strep | This study |
| pWL1564 | pGBDU-C1 | This study |
| pWL1565 | pGAD-C1 | This study |
| pWL1592 | pGBDU-C1-Mer2 | This study |
| pWL1593 | pGBDU-C1-Mer2-3A | This study |
| pWL1594 | pGBDU-C1-Mer2-4A | This study |
| pWL1596 | pGAD-C1-Mre11 | This study |
| pWL1753 | pGAD-C1-Spp1 | This study |
| pWL1598 | pGAD-C1-Rec102 | This study |
| pWL1597 | pGAD-C1-Rec104 | This study |
| pWL1625 | pGAD-C1-Ski8 | This study |
| pWL1595 | pGAD-C1-Rec114 | This study |
| pWL1629 | pGAD-C1-Spo11 | This study |

## Acknowledgements

We are extremely grateful to Francesca Mattiroli (Hubrecht Institute, Utrecht), for assistance with producing mononucleosomes. The histone H3 expression plasmid pET3 and the plasmid pUC19_601–147 DNA were a kind gift from Francesca Mattiroli. We are also grateful to Luke Berchowitz (Columbia University, New York), Andreas Hochwagen (NYU, New York), and Eric Greene (Columbia University, New York) for support, reagents, and expertise. The plasmid pUC18_601–167 DNA was a kind gift from Andrea Musacchio (MPI of Molecular Physiology, Dortmund). We thank Christopher Heim (MPI Developmental Biology) for technical help with SEC-MALS and MST measurements. We thank Andreas Blaha and Constanze Gremmelmaier for their technical support. Quantitative mass spectrometry and analysis was made possible by the EMBL Proteomics Core Facility (EMBL, Heidelberg). Work in the Weir lab is funded by the Max Planck Society and DFG grant WE 6513/2–1. Work in the Vader lab was funded by Max Planck Society and the European Research Council (ERC StG URDNA; agreement no. 638197), and an Amsterdam UMC Research Fellowship. SKF is funded by a studentship from the International Max Planck Research School (IMPRS) 'From Molecules to Organisms'.

## Additional information

### Funding

| Funder | Grant reference number | Author |
|---|---|---|
| Max-Planck-Gesellschaft | | Dorota Rousova<br>Vaishnavi Nivsarkar<br>Veronika Altmannova<br>Vivek B Raina<br>Saskia K Funk<br>David Liedtke<br>Petra Janning<br>Franziska Müller<br>Heidi Reichle<br>Gerben Vader<br>John Russell Weir |
| Deutsche Forschungsgemeinschaft | WE 6513/2-1 | Saskia K Funk<br>John Russell Weir |
| H2020 European Research Council | 638197 | Vaishnavi Nivsarkar<br>Vivek B Raina<br>Gerben Vader |
| Amsterdam UMC | Research Fellowship | Gerben Vader |

The funders had no role in study design, data collection and interpretation, or the decision to submit the work for publication.

### Author contributions

Dorota Rousová, Vaishnavi Nivsarkar, Conceptualization, Formal analysis, Investigation, Methodology; Veronika Altmannova, Formal analysis, Investigation, Methodology; Vivek B Raina, Formal analysis, Investigation; Saskia K Funk, Investigation, Methodology; David Liedtke, Investigation; Petra Janning, Formal analysis, Validation; Franziska Müller, Heidi Reichle, Methodology; Gerben Vader, Conceptualization, Funding acquisition, Investigation, Project administration, Supervision, Validation, Writing – original draft, Writing – review and editing; John R Weir, Conceptualization, Formal analysis, Funding acquisition, Investigation, Project administration, Supervision, Writing – original draft, Writing – review and editing

### Author ORCIDs

Dorota Rousová http://orcid.org/0000-0001-7395-2596
Vivek B Raina http://orcid.org/0000-0003-0507-9208
Gerben Vader http://orcid.org/0000-0001-5729-0991
John R Weir http://orcid.org/0000-0002-6904-0284

Decision letter and Author response
Decision letter https://doi.org/10.7554/eLife.72330.sa1
Author response https://doi.org/10.7554/eLife.72330.sa2

## Additional files

### Supplementary files
• Transparent reporting form
• Source data 1. Data files for XL-MS used to make the XL-MS maps in various figures.
• Source data 2. All IP-MS source data, used for quantitative mass-spec.

### Data availability
Source data has been provided for the XL-MS experiments and for the IP-mass spec data.

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
