## [Editor Report]

Using a combination of biochemical approaches and yeast genetics, the authors study the function of the DNA double-strand break factor Mer2. Rousova et al., show that Mer2 interacts with a meiotic chromosome axis factor (Hop1), nucleosomes, the nucleosome-binding protein Spp1, and the double-strand break factor Mre11 to serve as a "keystone" for meiotic DNA break formation. These findings represent an important step forward in understanding the functions of this highly conserved protein in meiosis.

---

## [Decision Letter]

**Decision letter after peer review:**

Thank you for submitting your article "Novel mechanistic insights into the role of Mer2 as the keystone of meiotic DNA break formation" for consideration by *eLife*. Your article has been reviewed by 3 peer reviewers, and the evaluation has been overseen by Federico Pelisch as the Reviewing Editor and Cynthia Wolberger as the Senior Editor. The following individual involved in review of your submission has agreed to reveal their identity: Owen Richard Davies (Reviewer #1).

Essential revisions:

All four Mer2 interactions studied here are of great interest but a deeper analysis will be required before publication. Further in vivo work may be the most promising in Mer2-Mre11 interactions, where some in vivo validation have already been done and therefore experiments would have a chance to close this topic rather than open up new experiments. Additionally, the interaction data for Mer2-Hop1 needs to be strengthen.

1) You will need to clarify what the logic was for choosing the selected combination of mutations in Mer2-3A and Mer2-4A. Are the mer3-3A and 4A mutations specifically disrupting the Mer2-mre11 interaction? The effect of the mutations on the Mer2-Mei4 and/or Mer2-Rec114 interactions should be addressed.

2) It should be assessed if Mer2-3A and 4A mutations have a strong effect on Mer2-Mre11 interaction in vivo. Co-IPs could be carried out, and/or localization of Mre11 to chromatin could be assessed by immunofluorescence.

3) As suggested in the reviews below, the pull-down data for the Mer2-Hop1 interaction would benefit from a 'clean-up' and a quantification of the signal from several repeats would be ideal. The same analysis should be applied to Figure 6b.

*Reviewer #1 (Recommendations for the authors):*

I have included below some comments and suggestions that could strengthen some aspects of the paper.

Line 117

Is it correct to call Mer2-Spp1 a constitutive complex when Spp1 is part of separate complexes outside of meiosis and that the authors could make the two proteins separately? I would reserve the term for situations in which the proteins require each other to fold are are seemingly always together in vivo.

Also, shouldn't Mer2-Spp1 be referred to as a 4:2 stoichiometry (rather than 2:4)?

Figure 1E

I would like to see these SEC-MALS data with the MWs plotted against a linear axis to assess how flat the MW traces are across each peak (the logarithmic scale can artificially flatten them when covering a large range). Any substantial sloping in the traces could indicate multiple species or dissociating species. Did the authors observe any evidence of this?

Figure 1F

The XL-MS data are intriguing and do question whether the Spp1 C-terminal coiled-coil is truly responsible for the interaction. Have the authors tested whether the interaction is retained upon deletion of Spp1's C-term, and whether an interaction is formed with only the C-term? This would be a straight-forward experiment and would strengthen this section of the manuscript (irrespective of the outcome).

Figure 3C

Were the SEC-MALS experiments performed using a dRI or UV measurement of mass concentration? If UV, how did the authors calibrate the parameters for analysing complex formation given that proteins would have different extinction coefficients? – this question applies to figures 1E and 3C. Also, if either dRI or UV, how were parameters calibrated for analysing Mer2-nucleosome complex formation in figure 3C (given the presence of DNA, these species would have different UV extinction coefficients and dn/dc values)? If they collected both UV and dRI signals, could they exploit these different values by using protein conjugate analysis to directly measure the mass ratio of Mer2 and nucleosomes?

The authors mention the 10 MDa species, but there also appears to be a species that elutes as a shoulder at the start of the 340 kDa peak that likely represents much more material than the 10 MDa peak – could they determine the MW of this peak and is it a multiple of the 340 kDa mass?

Also, did they run the fractions on a gel to confirm which peaks truly correspond to Mer2 and nucleosomes?

Figure 3F

There's a nice conserved patch at the C-term of Mer2 – ARTIIPWEEL – could this be responsible for nucleosome binding or does it have another known function? It would be nice to see the consequence of its deletion in the pulldown.

Line 290 – this should be Figure 3G.

Figure 5D (and in general)

I would be cautious about a structural interpretation of Mer2 as an anti-parallel tetrameric coiled-coil as it remains uncertain how the chains are arranged within its tetrameric core. Moreover, do we know that the core is a rigid linear coiled-coil or could it be a more highly bundled structure and/or with turns and flexibility? I assume this is based on Claeys Bouuaert et al., 2021? I checked this paper and the main biochemical evidence is from MALS analysis of a tethered dimer. However, these data appear relatively tentative as there was a 40% discrepancy between expected and observed mass, with signs of heterogeneity. The interpretation would require further biophysical analysis to confirm the presence of continuous helical structure, with solution particles consisting of the expected length and width. In absence of this (and assuming there are no other data that I am missing), the results could be artifacts of a more intricate, flexible and/or non-linear arrangement. Instead, I would simply define it as tetrameric, as this is irrefutable and will be consistent with either linear or non-linear future structures.

*Reviewer #2 (Recommendations for the authors):*

Overall, the study is well organized and clearly written, and follows a clear logical flow. Specific comments are noted below:

line 67 – "with" – should be "in" or "within"?

line 68 – "chromatin loop that emanate" should be "chromatin loops that emanate"?

Line 119: it would be helpful to note the extent of the predicted coiled-coil region when describing the four main Mer2 constructs used in the study. Is it the "core" region?

Lines 182-186: If the core region of Mer2 is indeed a tetrameric antiparallel coiled-coil, is it possible to model the register of this coiled-coil based on the crosslinking data?

Figure 1. E, Figure 3C: SEC-MALS data should include volume instead of time (flow rate is not known to the reader).

Line 203: reference figure 2A somewhere here.

Lines 254-262: It would be interesting (though not required for this paper) to see what would happen with Mer2 plus an array of nucleosomes. Would it phase-separate?

Figure 3E: At least two of these curves appear to be fit using a cooperative binding model. Do the authors feel that this is appropriate, given the multimeric nature of Mer2? (OK, now I see the note in Methods about this). The fitting mode should nonetheless be noted in the figure legend, and the Hill coefficient also noted in the figure or the legend.

Figure 3F – it would be helpful to have a graphical summary of these data, perhaps on a domain diagram of Mer2, to clarify which constructs of Mer2 do and do not interact with nucleosomes

Line 301: Reference for Stanzione is in superscript (also a problem in lines 495 and 552)

Line 314-315: The new Pch2/Hop1-related work from the San Segundo lab should also be cited here.

Figure 4C: This is a really interesting and well-designed set of experiments. However, the gel as shown is not really as convincing as the authors suggest in the text. In particular, the light bands in the pulldown lane for Step-Hop1-WT plus Mer2 raise some doubt on whether this is really showing "no interaction" or not. Perhaps these bands are contaminants, but in that case another set of lanes (maybe even in supplemental) showing that would be nice. Also, the label at right should read something like "Hop1 FL (WT or K593A)".

Lines 452-460: What about Rec114 and Mei4 as potential binding partners for the region of Mer2 interrupted by the 3A and 4A mutants? Are the authors confident that this interaction is not disrupted?

Given the ambiguity of the results in Mre11 direct pulldowns, and the importance/novelty of this result, I think it's important to establish that Mre11 is not properly recruited to DSB sites in the Mer2 3A and/or 4A mutants, perhaps by examining localization on chromosome spreads. Depending on the results, it may be important to also look for Rec114 and/or Mei4 in chromosome spreads.

Line 593: "shutting" should be "shutting down"?

Line 717: "Quadruplate" should be "Quadruplicate"

Given the complexity of the findings and the many functional domains of Mer2 now known, it would be very helpful if one of the later figures had a domain diagram of Mer2 with all of these regions, and their known or likely functions, highlighted.

Supp Figure 2 C and D, the description in page has a typo, the N terminus is a monomer and C terminus is a dimer.

*Reviewer #3 (Recommendations for the authors):*

The experiments are well executed, informative, and in most cases well explained, yet the manuscript appears unfinished and unfocused as it presents partially incomplete analysis of several distinct Mer2 interactions and functions. The manuscript examines the interaction of Mer2 with Spp1, nucleosomes, Hop1 and the Mre11 complex. Whereas Mer2-Spp1 interaction has been extensively studied and the Mer2-Mre11 interaction was also known, the Mer2-nucleosome and Mer2-Hop1 interactions were not described. The manuscript invests nearly the same effort, albeit with different approaches into all four interactions. Whereas this approach permits covering most Mer2 functions, increased emphasis on any one of the four stories would benefit the manuscript.

The Mer2-Spp1 interaction is biochemically characterized in the manuscript without functional validation in vivo, which is understandable given the extensive in vivo experiments that has published concerning the Mer2-Spp1 interaction.

The Mer2-nucleosome interaction is novel and is potentially very interesting for Mer2 function in DSB formation, yet only in vitro characterization is carried out without in vivo validation. Given that XL MS was used to map interacting residues between nucleosomes and Mer2 it would have been very revealing to test more systematically the relevance of the interacting residues both in vitro and in vivo.

The Mer2-Hop1 interaction was also not characterized earlier and the results suggest interesting regulation of Mer2-Hop1 interaction by Hop1 binding to Red1.It is proposed that Red1 binding to Hop1 disrupts an intramolecular interaction between Hop1 HORMA domain and a C terminal closure motif, which enables an interaction between Mer2 and the C terminus of Hop1. This hypothesis is tested by elegant experiments in vitro, but equivalent experiments are not carried out in vivo. An interesting experiment would be to target the C terminus of Hop1 to axial elements(e.g. by attaching it to Red1 as a tag) to test if Hop1 C terminus can support DSB formation by recruit Mer2 to axial elements. XL-MS experiments could have been used too to fine-map the interacting residues between Hop1 and Mer2 as prelude to design in vivo separation function mutants that disrupt Hop1-Mer2 interaction.

Finally, in the N terminus of Mer2, the manuscript identifies conserved residues that are important for Mer2-Mre11 interaction. Crucially, mutations of these residues abolish DSB formation in vivo. Despite the merits of these experiments, the analysis is incomplete.

First, it is not clearly explained what was the logic of choosing the selected combination of mutations in Mer2-3A and Mer2-4A. Did the authors generate other mutations in the N terminus, but only these had an effect? This information would be relevant to understand the specificity of the phenotypes.

Second, it is not clear if the mer3-3A and 4A mutations specifically disrupt only Mer2-mre11 interactions. Were the effect of these mutations tested on Mer2-Mei4 or Mer2-Rec114 interactions which are one of the most important interactions of Mer2?

Third, whereas the effect of mer2-3A and 4A is very severe on DSB formation, the interaction between Mer2 and Mre11 appears to be only moderately impaired in vitro by these mutations. Hence it is questionable if the moderate disruption of Mre11-Mer2 interaction can account for a complete loss of DSBs in the mer2-3A and 4A mutants.

a. Quantification should be provided for the in vitro effect, currently only a gel picture is shown.

b. Further, it should be assessed if Mer2-3A and 4A mutations have a strong effect on Mer2-Mre11 interaction in vivo. Co-IPs could be carried out, and/or localization of Mre11 to chromatin could be assessed by immunofluorescence. Even better, ChIP could be used to test if Mre11 is localized to DSB sites in mer2-3a and -4a mutants.

c. It should be also assessed if other components of the DSB machinery, e.g. Rec114 or Mei4 assemble on chromatin correctly in mer2-3a and -4a mutants.

Without addressing these questions the last project is not sufficiently rigorous.

In summary, the manuscript presents valuable and extensive datasets that would be worth publishing in *eLife*, but all the four stories of the manuscript are incomplete, hence further work would be necessary before publishing. In my opinion, it would be sufficient to develop one of the stories further, e.g. provide in vivo validation in story 2 or 3, or provide more evidence that the mer2-3a and 4a mutations specifically disrupt the Mer2-Mre11 interaction in vivo. The authors could be given the option to develop further any one of the four stories(preferentially the last three as they are the most novel) along the lines suggested above.

---

## [Author Response]

Essential revisions:All four Mer2 interactions studied here are of great interest but a deeper analysis will be required before publication. Further in vivo work may be the most promising in Mer2-Mre11 interactions, where some in vivo validation have already been done and therefore experiments would have a chance to close this topic rather than open up new experiments. Additionally, the interaction data for Mer2-Hop1 needs to be strengthen.1) You will need to clarify what the logic was for choosing the selected combination of mutations in Mer2-3A and Mer2-4A. Are the mer3-3A and 4A mutations specifically disrupting the Mer2-mre11 interaction? The effect of the mutations on the Mer2-Mei4 and/or Mer2-Rec114 interactions should be addressed.

We have clarified our logic for the 3A and 4A mutants in the text. Briefly, we searched for mutants that would disrupt interactions dependent upon the N-terminal 139 amino acids of Mer2 and identified one conserved region between amino acids 52 and 72 (see Figure 5A). We made two mutants in this region, one mutating a highly-conserved core of three amino acids (WxxKxxL) to alanine (AxxAxxA – 3A) and a second region of four less well conserved amino acids, also to alanine (4A). The logic of the 4A mutant was to mitigate a possible outcome in which the 3A mutant would give rise to an unstable (recombinant) protein. This was not the case, and as such we have been able to test both alleles: we included data on 4A throughout the manuscript for the sake of completeness.

We attempted to address the effect of Mer2^3A^ and Mer2^4A^ on Mei4 and Rec114 interaction using three approaches. Firstly using Y2H, but high background self-activation unfortunately precluded us from pursuing this avenue (see Figure 6—figure supplement 1). Secondly, we carried out co-IPs between 13MYC-tagged Mer2 (wild type, 3a and 4a) and 3HA-Rec114. Using this approach we however did not detect an interaction between these two factors. This data is now part of Figure 6—figure supplement 2. Thirdly, we carried out an IP-mass spec of Mer2^WT^ versus Mer2^4A^. While we detected several known protein interactors of Mer2 using this approach (e.g. Spp1, and nucleosomal Histones), we did not detect an interaction between Mer2 and Rec114 (see Figure 5F and Figure 5—figure supplement 2). Finally, we carried out immunofluorescence (IF) on chromosomal spreads to investigate any possible chromosome association effects on Rec114 (and also Mre11) in the Mer2 mutant background. We observed no obvious disruption of the localisation of Mre11 or Rec114 in the mutant versus the wildtype (Figure 6—figure supplement 3). We conclude that the interaction between both Mer2 and Mre11 and Mer2 and Rec114 is either transient, represents only a small fraction of the cellular population of Mer2, or is mediated by a mechanism that is not compatible with the approaches we have taken here. Furthermore, our IF analysis suggests that the introduction of the Mer2^3A^/Mer2^4A^ mutations, while leading to strong DSB defects, does not lead to an observable defect in the recruitment of Rec114/Mre11. These observations argue that interference with the binding between Mre11 and Mer2 does not disrupt the DSB machinery per se, but rather causes a more specific, further to be characterized, biochemical defect.

To alleviate the reviewers’ concerns regarding the specificity of the Mer2-3A (and 4A) mutants, we examined the interaction with Hop1, which should not be disrupted by the mutation. In both the Mer2 co-IPs Figure 6—figure supplement 2, and the Mer2 IP-MS (Figure 5G) we could detect robust interaction for both Mer2-WT and Mer2-3A (and Mer2-4A in the case of the co-IPs) with Hop1.

2) It should be assessed if Mer2-3A and 4A mutations have a strong effect on Mer2-Mre11 interaction in vivo. Co-IPs could be carried out, and/or localization of Mre11 to chromatin could be assessed by immunofluorescence.

We have answered this question in detail above, and refer the reviewer to those comments. We are only able to detect an in vivo interaction between Mer2 and Mre11 using Y2H. However in order to shed more light on the interaction between Mer2 and Mre11 we carried out XL-MS on this complex. The results are now included in figure 6D. In addition we analysed the chromosomal localisation of Mre11 was apparently unaffected by the mutants (Figure 6—figure supplement 3). We conclude from this, as also discussed in the manuscript, that our Mer2 mutants interfere potentially with an allosteric effect and not with the recruitment of Mre11 to chromosomes.

3) As suggested in the reviews below, the pull-down data for the Mer2-Hop1 interaction would benefit from a 'clean-up' and a quantification of the signal from several repeats would be ideal. The same analysis should be applied to Figure 6b.

We have repeated all the pulldown experiments, and quantitated the results. In the case of Hop1 we carried out quadruplicate experiments using a modified protocol to clarify the experiment. In this case we were able to quantitate binding. Under these conditions we also modified our Mer2-Hop1 binding model. In this case we do observe binding between Hop1-WT and Mer2 that is increased in Hop1-K593A mutant. We also do not see an interaction between Mer2 and the Hop1∆HORMA (Figure 4C and D). As such we consider that the effect on the Hop1-Mer2 interaction in the presence of Red1 (Figure 4A and B) is more nuanced than simply through the displacement of the C-terminus of Hop1 from interaction with the HORMA domain. We have addressed this in the discussion.

Reviewer #1 (Recommendations for the authors):I have included below some comments and suggestions that could strengthen some aspects of the paper.Line 117Is it correct to call Mer2-Spp1 a constitutive complex when Spp1 is part of separate complexes outside of meiosis and that the authors could make the two proteins separately? I would reserve the term for situations in which the proteins require each other to fold are are seemingly always together in vivo.

We have removed the term “constitutive” when discussing Mer2-Spp1

Also, shouldn't Mer2-Spp1 be referred to as a 4:2 stoichiometry (rather than 2:4)?

We have corrected this.

Figure 1EI would like to see these SEC-MALS data with the MWs plotted against a linear axis to assess how flat the MW traces are across each peak (the logarithmic scale can artificially flatten them when covering a large range). Any substantial sloping in the traces could indicate multiple species or dissociating species. Did the authors observe any evidence of this?

We thank the reviewer for this suggestion. We have altered the axes on this, and on other plots, to show the elution volume (rather than time) on the X-axis, and the linear MW on the Y axis, both in response to this comment, and the other reviewer comments (see below). We do not see evidence of dissociation, rather the presence of some higher molecular weight species, in particular in the presence of Mer2.

Figure 1FThe XL-MS data are intriguing and do question whether the Spp1 C-terminal coiled-coil is truly responsible for the interaction. Have the authors tested whether the interaction is retained upon deletion of Spp1's C-term, and whether an interaction is formed with only the C-term? This would be a straight-forward experiment and would strengthen this section of the manuscript (irrespective of the outcome).

We have carried out this experiment using two constructs of Spp1, one containing amino acids 1-170 (PHD domain) and the other containing amino acids 169-353 (coiled-coil domain). We found that Spp1 1-170 (referred to as Spp1∆C in the revised manuscript) did not bind to Mer2, but that Spp1 169-353 (referred to as ∆PHD in the manuscript) did bind to Mer2. We also tried making an Spp1 lacking amino acids 249-353, to further refine the role of the C-terminus. However in our hands this construct was unstable and thus results on Mer2 binding were uninterpretable. We have included this data as a new Figure 1—figure supplement 2. We have also attempted to create a model of Mer2 with Spp1 using AlphaFold2 multimer, but the algorithm reported very poor predicted aligned error scores between Mer2 and Spp1.

Figure 3CWere the SEC-MALS experiments performed using a dRI or UV measurement of mass concentration? If UV, how did the authors calibrate the parameters for analysing complex formation (given that proteins would have different extinction coefficients?) – this question applies to figures 1E and 3C. Also, if either dRI or UV, how were parameters calibrated for analysing Mer2-nucleosome complex formation in figure 3C (given the presence of DNA, these species would have different UV extinction coefficients and dn/dc values)? If they collected both UV and dRI signals, could they exploit these different values by using protein conjugate analysis to directly measure the mass ratio of Mer2 and nucleosomes?

We have used dRI in all experiments, and have now made this clear in the methods section. We have considered what the reviewer is asking, and have consulted with other biochemists working in nucleosomes. We argue that the DNA signal would dominate and that the analysis of the UV/dRI signal might not be quantitative. Reading between the lines, we reconsider the question – are we sure of the nucleosome Mer2 complex stoichiometry? In this case, previous SEC-MALS analysis made by Karolin Luger and coworkers on nucleosomes and nucleosome protein complexes (Gaullier et al., PLoS ONE 2020) treated the measured molecular mass the same for all samples. Importantly, the measured mass for nucleosomes showed less discrepancy than for the protein complexes. As such, we are confident that we do not need to take into account the difference in light scattering contributed by DNA. Likewise using a mass photometry in Figure 3B, we observe only a 7% discrepancy in molecular mass determination for nucleosomes alone, despite calibration being done using protein standards. As such, we consider that the majority of the species we observe correspond to 1:1 complexes of Mer2 tetramer to nucleosome.

The authors mention the 10 MDa species, but there also appears to be a species that elutes as a shoulder at the start of the 340 kDa peak that likely represents much more material than the 10 MDa peak – could they determine the MW of this peak and is it a multiple of the 340 kDa mass?

We have re-analysed this peak and this has now been included in an extra supplementary figure (Figure 3—figure supplement 1) where the Y axis also shows the linear mass. We see that the shoulder is a mixture of species with an average molecular mass of 1.45 MDa; this would make it closest to a tetramer of Mer2-mononucleosome complexes. We would like to currently refrain from further speculation about what the significance of this- if any, or what the 10 MDa species might be. We consider that this might be best achieved in a dedicated study using a combination of SEC-MALS, SEC-SAXS, EM and molecular modelling to understand the potential meaning of these higher order species.

Also, did they run the fractions on a gel to confirm which peaks truly correspond to Mer2 and nucleosomes?

Unfortunately, due to the experimental setup the fraction collector is not cooled. As such by the time we came to run fractions on a gel, we had significant problems with sample degradation.

Figure 3FThere's a nice conserved patch at the C-term of Mer2 – ARTIIPWEEL – could this be responsible for nucleosome binding or does it have another known function? It would be nice to see the consequence of its deletion in the pulldown.

We already see that Mer2∆C does not bind to nucleosomes (Figure 3F lane 2) in a pulldown experiment with biotinylated nucleosomes. The reviewer however does raise an interesting point and we are continuing to investigate the role of this motif. Characterizing this domain in detail warrants a significant time investment and we feel that such an investment currently falls beyond the scope of our manuscript.

Line 290 – this should be Figure 3G.

This has been resolved.

Figure 5D (and in general)I would be cautious about a structural interpretation of Mer2 as an anti-parallel tetrameric coiled-coil as it remains uncertain how the chains are arranged within its tetrameric core. Moreover, do we know that the core is a rigid linear coiled-coil or could it be a more highly bundled structure and/or with turns and flexibility? I assume this is based on Claeys Bouuaert et al., 2021? I checked this paper and the main biochemical evidence is from MALS analysis of a tethered dimer. However, these data appear relatively tentative as there was a 40% discrepancy between expected and observed mass, with signs of heterogeneity. The interpretation would require further biophysical analysis to confirm the presence of continuous helical structure, with solution particles consisting of the expected length and width. In absence of this (and assuming there are no other data that I am missing), the results could be artifacts of a more intricate, flexible and/or non-linear arrangement. Instead, I would simply define it as tetrameric, as this is irrefutable and will be consistent with either linear or non-linear future structures.

The reviewer is correct, our description of the anti-parallel coiled-coils is based on the Claeys Bouuaert et al., 2021 Nature paper. In line with the reviewer’s suspicions, modelling of the central coiled coil using AlphaFold2 Multimer consistently models the region as a parallel arrangement of coiled-coils with a moderate level of confidence. However, plotting the cross-linking data onto this model (even taking the highest confidence cross-links with the most observations) the result is ambiguous. Some cross-links are entirely constituent with the model, whereas others are too distant. As such we concur with the reviewer’s suggestion that Mer2 might have an unconventional structural arrangement. We have limited speculation on Mer2’s structure in the manuscript so as not to mislead the reader.

Reviewer #2 (Recommendations for the authors):Overall, the study is well organized and clearly written, and follows a clear logical flow. Specific comments are noted below:line 67 – "with" – should be "in" or "within"?line 68 – "chromatin loop that emanate" should be "chromatin loops that emanate"?

Corrected

Line 119: it would be helpful to note the extent of the predicted coiled-coil region when describing the four main Mer2 constructs used in the study. Is it the "core" region?

Yes, the “core” region corresponds to the predicted coiled-coil region. We have clarified that in the manuscript, and also annotated the domain cartoons in Figure 1 with coiled-coil prediction.

Lines 182-186: If the core region of Mer2 is indeed a tetrameric antiparallel coiled-coil, is it possible to model the register of this coiled-coil based on the crosslinking data?

In response to Reviewer 1’s input (see above) we have revised the description of the crosslinking data.

Figure 1. E, Figure 3C: SEC-MALS data should include volume instead of time (flow rate is not known to the reader).

We have corrected this: all SEC-MALS experiments now include volume on the X-axis. This is particularly relevant to Figure 1E, where MBP-Mer2-core has a similar elution volume as MBP-Spp1 (though different molecular masses).

Line 203: reference figure 2A somewhere here.

We have added references to both Figure 2A and 2B.

Lines 254-262: It would be interesting (though not required for this paper) to see what would happen with Mer2 plus an array of nucleosomes. Would it phase-separate?

It is an intriguing and important experiment, and we do intend to look at this. To give a more verbose answer, we are of the opinion that the possible (presumed) phase-separation of Mer2 (together with Rec114 and Mei4) would be best tested in the context of not only chromatinised DNA, but by using a (partial) reconstitution of the loop-axis architecture, making use of Red1-Hop1 components (since we know Mer2 binds to Hop1 in the context of Red1).

Figure 3E: At least two of these curves appear to be fit using a cooperative binding model. Do the authors feel that this is appropriate, given the multimeric nature of Mer2? (OK, now I see the note in Methods about this). The fitting mode should nonetheless be noted in the figure legend, and the Hill coefficient also noted in the figure or the legend.

We have included the Hill coefficient in the figure legend.

Figure 3F – it would be helpful to have a graphical summary of these data, perhaps on a domain diagram of Mer2, to clarify which constructs of Mer2 do and do not interact with nucleosomes

We have included this as Figure 3G also integrating the EMSA data from Figure 3—figure supplement 2.

Line 301: Reference for Stanzione is in superscript (also a problem in lines 495 and 552)

Corrected

Line 314-315: The new Pch2/Hop1-related work from the San Segundo lab should also be cited here.

Added.

Figure 4C: This is a really interesting and well-designed set of experiments. However, the gel as shown is not really as convincing as the authors suggest in the text. In particular, the light bands in the pulldown lane for Step-Hop1-WT plus Mer2 raise some doubt on whether this is really showing "no interaction" or not. Perhaps these bands are contaminants, but in that case another set of lanes (maybe even in supplemental) showing that would be nice. Also, the label at right should read something like "Hop1 FL (WT or K593A)".Lines 452-460: What about Rec114 and Mei4 as potential binding partners for the region of Mer2 interrupted by the 3A and 4A mutants? Are the authors confident that this interaction is not disrupted?

The reviewer raises an important point, but there is difficulty in carrying out this experiment. Put simply, we do not observe a direct interaction between Mer2 and Rec114/Mei4 in a conventional in vitro experiment. Likewise, despite having a high quality Rec114-Mei4 sample (something that we have struggled to obtain), Claeys Bouuaert et al., 2021 only reported an interaction between Mer2 and the Rec114/Mei4 complex when they phase separated together on DNA and in the presence of a crowding agent. In agreement with this, in co-IP and IP-MS we did not observe an interaction between Mer2-WT and Rec114 (nor for Mei4 in the case of the mass-spec) (see also above). In chromosomal spreads of meiotic cells we see did not observe a significant disruption of the localisation of Rec114 to chromatin (Figure 6—figure supplement 3). Thus, in the absence of a positive control (Mer2-WT binding to Rec114-Mei4) we are unable to exclude the possibility that Mer2-3A/4A might also perturb binding to this complex. We have therefore included a statement in our discussion with regards to the additional interactions that have been previously described between Mer2 and factors in the DSB machinery (Arora et al., 2004)

Given the ambiguity of the results in Mre11 direct pulldowns, and the importance/novelty of this result, I think it's important to establish that Mre11 is not properly recruited to DSB sites in the Mer2 3A and/or 4A mutants, perhaps by examining localization on chromosome spreads. Depending on the results, it may be important to also look for Rec114 and/or Mei4 in chromosome spreads.

We have carried out extensive work to address this, and discuss in depth in comments to the editor, and we refer the reviewer to those comments.

Line 593: "shutting" should be "shutting down"?

Corrected.

Line 717: "Quadruplate" should be "Quadruplicate"

Corrected.

Given the complexity of the findings and the many functional domains of Mer2 now known, it would be very helpful if one of the later figures had a domain diagram of Mer2 with all of these regions, and their known or likely functions, highlighted.

We concur, especially in light of the additional XL-MS data. We have created two additional figures. In Figure 6F we have added a domain cartoon of Mer2 showing the overview of presumed interactions. In Figure 6—figure supplement 3 we have tried to integrate the observations from the cross-linking data into a single representation to highlight in detail those regions of Mer2 that are proximal to their binding partners, and vice-versa on the domains of Spp1, Nucleosomes, Hop1 and Mre11.

Supp Figure 2 C and D, the description in page has a typo, the N terminus is a monomer and C terminus is a dimer.

The reviewer is correct and the descriptions of the truncations used in the supplementary figures were confusing and we have clarified this.

Reviewer #3 (Recommendations for the authors):The experiments are well executed, informative, and in most cases well explained, yet the manuscript appears unfinished and unfocused as it presents partially incomplete analysis of several distinct Mer2 interactions and functions. The manuscript examines the interaction of Mer2 with Spp1, nucleosomes, Hop1 and the Mre11 complex. Whereas Mer2-Spp1 interaction has been extensively studied and the Mer2-Mre11 interaction was also known, the Mer2-nucleosome and Mer2-Hop1 interactions were not described. The manuscript invests nearly the same effort, albeit with different approaches into all four interactions. Whereas this approach permits covering most Mer2 functions, increased emphasis on any one of the four stories would benefit the manuscript.The Mer2-Spp1 interaction is biochemically characterized in the manuscript without functional validation in vivo, which is understandable given the extensive in vivo experiments that has published concerning the Mer2-Spp1 interaction.The Mer2-nucleosome interaction is novel and is potentially very interesting for Mer2 function in DSB formation, yet only in vitro characterization is carried out without in vivo validation. Given that XL MS was used to map interacting residues between nucleosomes and Mer2 it would have been very revealing to test more systematically the relevance of the interacting residues both in vitro and in vivo.

We have carried out IP-MS on Mer2, and compared this to the 4A mutant. In line with the EMSAs (now Figure 3—figure supplement 2C) we observe a slightly weaker interaction between Mer2 and histones (Figure 5G). We do agree with the reviewer that the interaction between Mer2 and nucleosomes is interesting, but that the mode of interaction is currently somewhat enigmatic. We consider that this would be best addressed through structural studies on complexes of Mer2 on nucleosomes, and this is the subject of ongoing work in our laboratory. In light of a Mer2-Nucleosome structure we would then be in a stronger position to make more penetrant mutants in vivo, and use those to explore their effects on Mer2 localization, co-factor localization, and DSB formation.

The Mer2-Hop1 interaction was also not characterized earlier and the results suggest interesting regulation of Mer2-Hop1 interaction by Hop1 binding to Red1.It is proposed that Red1 binding to Hop1 disrupts an intramolecular interaction between Hop1 HORMA domain and a C terminal closure motif, which enables an interaction between Mer2 and the C terminus of Hop1. This hypothesis is tested by elegant experiments in vitro, but equivalent experiments are not carried out in vivo. An interesting experiment would be to target the C terminus of Hop1 to axial elements(e.g. by attaching it to Red1 as a tag) to test if Hop1 C terminus can support DSB formation by recruit Mer2 to axial elements. XL-MS experiments could have been used too to fine-map the interacting residues between Hop1 and Mer2 as prelude to design in vivo separation function mutants that disrupt Hop1-Mer2 interaction.

We have carried out the XL-MS experiments on the Mer2-Hop1 complex as suggested by the reviewer. This data is now found in Figure 4F. Further analysis of mutations in both Hop1 and Mer2, and in their orthologs, is the subject of further work in our lab, and the in vivo analysis is work that we are pursuing with collaborators.

Finally, in the N terminus of Mer2, the manuscript identifies conserved residues that are important for Mer2-Mre11 interaction. Crucially, mutations of these residues abolish DSB formation in vivo. Despite the merits of these experiments, the analysis is incomplete.First, it is not clearly explained what was the logic of choosing the selected combination of mutations in Mer2-3A and Mer2-4A. Did the authors generate other mutations in the N terminus, but only these had an effect? This information would be relevant to understand the specificity of the phenotypes.

We have explained the logic of our choices (see above in our response to comments from the editor).

Second, it is not clear if the mer3-3A and 4A mutations specifically disrupt only Mer2-mre11 interactions. Were the effect of these mutations tested on Mer2-Mei4 or Mer2-Rec114 interactions which are one of the most important interactions of Mer2?

Again, we have tried to address this exhaustively. We are unable to detect interaction with Rec114 and Mei4 by co-IPs, chromosomal spreads followed by IF, nor by IP followed by mass-spec. We do, however, see robust interaction between Mer2 and Hop1, and Mer2-3A and Mer2-4A with Hop1.

Third, whereas the effect of mer2-3A and 4A is very severe on DSB formation, the interaction between Mer2 and Mre11 appears to be only moderately impaired in vitro by these mutations. Hence it is questionable if the moderate disruption of Mre11-Mer2 interaction can account for a complete loss of DSBs in the mer2-3A and 4A mutants.a. Quantification should be provided for the in vitro effect, currently only a gel picture is shown.

We have carried out replicates of the pulldown analysis between Mer2, Mer2-3A, Mer2-4A and Mre11. We concur that the in vitro effect is not terribly strong, despite the fact that it is robust. However, we observe a severe interaction effect in Y2H analysis. Additionally, we observe both in EMSAs and in quantitative IP-MS analysis a mild reduction in chromatin (nucleosome) binding. As such, we have added the possibility in our discussion that the strong effect on DSB formation could arise through a combinatorial effect, for example (also based on insights gathered from additional XL-MS data) we explicitly discuss a potential role of the SUMOylation on Mer2 and the SUMO interaction motif in the C-terminus of Mre11 in promoting the binding of Mer2 to Mre11.

b. Further, it should be assessed if Mer2-3A and 4A mutations have a strong effect on Mer2-Mre11 interaction in vivo. Co-IPs could be carried out, and/or localization of Mre11 to chromatin could be assessed by immunofluorescence. Even better, ChIP could be used to test if Mre11 is localized to DSB sites in mer2-3a and -4a mutants.

As described above we have performed these experiments, but did not detect interaction between Mer2 and Mre11 using three different in vivo approaches. These results are discussed in detail in the comments to the editor.

c. It should be also assessed if other components of the DSB machinery, e.g. Rec114 or Mei4 assemble on chromatin correctly in mer2-3a and -4a mutants.Without addressing these questions the last project is not sufficiently rigorous.

As described above, we have carried out these experiments. We analysed the chromosomal recruitment of Rec114 (and Mre11) in cells expressing MER2, mer2-3a, mer2-4a. We found that Rec114 (and Mre11) appear to be recruited to chromosomes in all three backgrounds. These data are now presented in Figure 6—figure supplement 3.

In summary, the manuscript presents valuable and extensive datasets that would be worth publishing in eLife, but all the four stories of the manuscript are incomplete, hence further work would be necessary before publishing. In my opinion, it would be sufficient to develop one of the stories further, e.g. provide in vivo validation in story 2 or 3, or provide more evidence that the mer2-3a and 4a mutations specifically disrupt the Mer2-Mre11 interaction in vivo. The authors could be given the option to develop further any one of the four stories(preferentially the last three as they are the most novel) along the lines suggested above.

We appreciate the reviewer’s concerns. Although our attempts to carry out further in vivo techniques were not entirely successful (see above), we have included quantitative mass-spec analysis of IPs from Mer2-WT and Mer2-3A. We have also obtained and added XL-MS data for both Hop1-Mer2 and the Mre11-Mer2 complexes, and used existing structural information/modelling to provide additional information as to how Mer2 might interact with both of these proteins, gaining additional insights. We have improved our biochemical data through quantitative analysis of the Hop1 and Mre11 pulldowns. As such, we consider that we have developed all four of our stories further, especially with regards to the biochemical data and XL-MS maps.